# Performance Evaluation of Clay Plastic Concrete of Cement and Epoxy Resin Composite as a Sustainable Construction Material in the Durability Process

Sanaz Soltaninejad [1] , Seyed Morteza Marandi [1,*] and Naveen BP [2,*]

1. Department of Civil Engineering, Shahid Bahonar University of Kerman, Kerman 7616913439, Iran; s.soltaninejad@eng.uk.ac.ir
2. Department of Civil Engineering, Amity University Haryana, Gurugram 122412, India
* Correspondence: marandi@uk.ac.ir (S.M.M.); bpnaveen@ggn.amity.edu (N.B.); Tel.: +98-9131404195 (S.M.M.); +91-9916232349 (N.B.)

**Abstract:** In this research, bentonite soil was stabilized with cement and epoxy resin additives by gradually increasing the ratio of epoxy resin to water to withstand six successive wetting/drying (W/D) cycles. Unconfined compressive strength (UCS) tests were performed on the stabilized samples after curing and 24 h of soaking in the third and sixth cycles. The swelling–shrinkage potential of bentonite soil was evaluated indirectly by analyzing its stress–strain behavior. The results showed that for different amounts of cement, the minimum ratio of epoxy resin to water added to bentonite soil to last up to six cycles was equal to 1:1. Also, by stabilizing the bentonite soil so that the total weight of the optimum moisture content was replaced with epoxy resin, the strength and rigidity attained the level of normal concrete, with the difference that the ductility was much more significant. The failure strain value increased by 32 times, and the plastic region of the stress–strain curve expanded over the wide strain interval with a length of about 5%. Clay plastic concrete design consisting of cement and epoxy resin develops sustainable ground improvement methods.

**Keywords:** clay plastic concrete; sustainable construction material; swelling–shrinkage; wetting-drying cycles; epoxy resin; toughness; soil stabilization

## 1. Introduction

One of the major problems of structural foundations, specifically for light structures, is the presence of expansive soils. These soils are usually composed of clay minerals such as montmorillonite, illite, and kaolinite with high-plasticity properties. The expansive soils increase in volume with water absorption and conversely shrink as they dry. The volume changes of these soils, which are due to their moisture changes, cause extensive damage to many road construction projects, buildings, bridges, and other structures placed on them [1,2]. In the United States, the value of damages due to expansive soils sums to about USD 2.3 billion annually. This exceeds the average annual costs of flood, hurricane, and earthquake damages [3,4].

Different types of failures seen in road pavement on swelling soils include grooves, cracks, depressions, fragmentation, polishing, roughness, etc. These cracks allow water to enter the road structure, and eventually, it leads the roads to fail. Construction of rigid pavement will not be successful due to cracking created by swelling of the subgrade soil. However, constructing a flexible pavement also requires the effective impermeability of the base and subbase layers, accomplished via subsurface membranes [5].

Increases in the moisture of expansive soils occur due to rain falling, frosting, melting snow and ice in the hot season, rising groundwater levels, cracking in water pipes, sewage disposal system or water drainage, etc. The shrinkage is the reflection of the expansion [3].

Swelling soils increase in volume and enter a sticky state after absorbing water. Conversely, such soils become extremely hard by drying, and block cracks are created on the ground surface. The cyclic movements of the foundation, including going up and down, engendering cracks and burnout in the structures [2,6]. This dual behavior of soil, which becomes viscous and increases in volume by absorbing water and shrinks and hardens by losing water, is called "swelling-shrinkage" behavior. The primary reason for the extensive damage caused by expansive soils Is the swelling–shrinkage behavior of soil [7,8].

One of the main concerns of engineers is to focus on soil behavior in two steps, namely, the construction and design of a structure for longevity. Expansive soils have a high potential for swelling that produces significant volume changes and lifting forces. Therefore, it will be crucial to consider both swelling and the strength behaviors of the expansive soils as the main design factor in constructing a road [9]. The critical point about expansive soils is first identifying and then correctly estimating the swelling amount due to the changes in the environment to provide suitable options for improving and controlling the volume change [10]. In a reconstruction of the damages caused by the swelling of road subgrade soil, if the breakdown factor is not remedied after the restoration of the road surface, the possibility of re-rupture is not far from the mind [11].

Due to the problems that the presence of swelling soils imposes on engineering structures and also their widespread distribution around the world, it is necessary to conduct studies to evaluate their swelling potential along with engineering mineralogy. This will reduce their problems and economic costs to the minimum. Therefore, stabilization of these types of soils using additives is essential. Different types of additives are used for soil stabilization. The choice of each additive depends on various factors such as soil type, weather conditions, the purpose of stabilization, and environmental and economic issues. The chemical additives are those stabilizing agents that react chemically with soil minerals and change the structure [12]. Portland cement and lime are commonly used members of this group of additives. Adhesive additives lead the soil particles to stick to each other without reacting to them. They increase the strength and alter the compaction properties and the sensitivity to moisture.

Research has been conducted on the use of solutions for binding soil particles such as resins and polymers that have significant impacts on improving the physical and mechanical properties of sandy soils [13–17], fine-grained soils (silt and clay) [18–20], and dispersive soils [21]. Other studies have also employed simultaneously cement and adhesive additive in the form of two-component water-soluble epoxy resins including resin and hardener to stabilize clay [22–24] and silty clay [25].

The review history shows that different researchers have proposed various methods of both laboratory and field research to identify expansive soils and measure their swelling–shrinkage pressure. In the most common methods, odometer tests are used [26,27]. The potential of soils for volume changes directly depends on the activity amount of their dominant clay minerals and, in other words, their plasticity [8,28–31].

Concrete is one of the most common construction materials used in civil engineering projects. The ever-growing using of cement in the construction industry has led to an increase in cement production [32,33]. In order to produce one ton of cement, a huge amount of thermal energy is needed, and as a result, approximately one ton of $CO_2$ is emitted. Cement production plays an important role in global warming and increasing $CO_2$ emissions [34–37]. Therefore, the use of ordinary Portland cement is associated with severe environmental concerns. All international communities are trying to address the causes of global warming and the harmful effects of the increase in the Earth's temperature. Additionally, the environmental taxes that arose after the adoption of the Kyoto Protocol [38] caused the substitution of other harmless substances to be researched [36].

The use of natural materials with little environmental effect which improve the mechanical characteristics of building materials will have remarkable advantages for sustainable construction [33]. Replacing part of the cement in concrete with natural or industrial mineral additives plays an important role in producing more environmentally friendly

building materials and achieving sustainability targets [33]. It has even been recommended in some studies to use natural materials to achieve sustainable concrete that is more efficient [39–42]. The mechanical performance and durability of concrete with recycled fine/coarse aggregates due to its high content of hardened old mortar are lower than those of concrete with natural aggregates [43]. Wu et al. [44] investigated the micro and macro characteristics of sustainable concrete, in which a mixture of waste concrete fine (WCF) and waste brick fine (WBF) was used as a substitute for cement and sand. According to their results, the properties of sustainable concrete that uses the WCF/WBF combination to replace both 10% of cement content and 10% of sand content are similar to or better than conventional concrete.

Clay minerals such as metakaolin, kaolin, and montmorillonite are used in concrete production due to their high potential in improving hydration reactions and reducing clinker agent amounts, energy consumption, and greenhouse gas emissions. As a result, they reduce cement consumption [45–48]. Extensive research has been performed on the use of Kaolin in the cement and concrete industries. Bentonite clay is less commonly used as a partial replacement for cementitious materials due to its low pozzolanic activity, high hydrophilicity, and significant swelling nature [49,50].

Bentonite clay in a study by Gedik and Atmaca [33] was replaced from 0 to 30% by volume with Portland cement CEM II 42.5R. The compressive strength of the 28-day stabilized samples with 5% bentonite clay increased by 94.7%, and their thermal conductivity decreased by 31.2%. In addition, if the bentonite clay content of the samples was more than 15%, their mechanical and thermal conductivity characteristics considerably reduced. In another study, bentonite soil was examined as a partial replacement for cement utilization. The ratio of bentonite and other natural substances (kaolin and resin) was variable. The strength parameter (compressive and tensile) of 18 mixing combinations was determined. The results showed that using bentonite, kaolin, and resin at a 2.7% ratio by weight of the cement increased the UCS by approximately 6%. Without cement, the presence of bentonite and other natural materials in the concrete mixing plan was useless [36].

This research was conducted based on two goals: (A) to indirectly evaluate the swelling-shrinking potential of bentonite clay, and (B) to examine the use of stabilized bentonite clay as an alternative to ordinary Portland cement (OPC) concrete from an environmental and sustainability point of view. Microstructural studies were performed on bentonite clay soil. It was initially stabilized with cement to undergo the durability process for the initial assessment of the swelling–shrinkage potential. Each cycle consisted of 24 h of the sample being immersed in water and then 24 h of drying in an oven at 70 °C. The swelling–shrinkage potential of bentonite soil was such that despite stabilization with the highest amount of cement and long-term curing for 28 days, it lasted for fewer than two cycles of W/D.

Checking the review history related to objective (A) of the research shows that no studies have been conducted to stabilize bentonite soil to last against successive W/D cycles so far. In order to overcome the swelling–shrinkage potential of bentonite soil during the W/D cycles, a traditional additive such as cement alone was not sufficient. It was necessary to increase the strength and rigidity to high levels. Therefore, an epoxy resin additive was employed along with cement to stabilize the bentonite soil at a level that withstood six successive cycles of W/D. The long-term curing time for improving the strength parameter values of the stabilized bentonite soil samples with cement was not effective. On the other hand, the clogging of epoxy resin is in the early hours. So, the stabilized samples with 4, 8, and 12% cement and epoxy resin-to-water ratios of 0.25, 0.5, 1, and 2 were treated for 7 days. They were then subjected to the durability process.

The plastic region of the stress–strain curve of bentonite soil stabilized with only epoxy resin extended over the wide strain interval with a length of approximately 5%. In the plastic zone, the stress was almost within the range of OPC concrete's strength. The combination of bentonite soil and epoxy resin formed concrete material, for which, unlike OPC concrete, the failure strain and toughness increased by several times as the strength

increased. This concrete material composed of bentonite and epoxy resin was named clay plastic concrete [23,24,51]. In addition to its remarkable strength, clay plastic concrete had considerably more ductility and toughness against failure than OPC concrete. In addition, clay plastic concrete was lighter than OPC concrete.

A review of the research history related to objective (B) reveals that the concrete industry is the largest consumer of cement and aggregates. Research on concrete without cement and aggregates with similar or better performance than OPC concrete is a big step toward saving energy and promoting sustainability in the construction industry. The use of bentonite clay concrete, which has significant plastic properties, as a replacement for OPC concrete, which consumes about 300 to 500 kg of cement per m$^3$ of concrete [52,53], could allow sustainable development and sustainability. Environmental damage could be minimized by clay plastic concrete. Therefore, it will contribute to remarkable progress toward sustainable construction. Clay plastic concrete design using bentonite clay could be an introduction to green building design using sustainable materials.

## 2. Materials and Methods

### 2.1. Materials

The bentonite soil was a cream-colored powder prepared by the Farzan Powder Mining Company located in Khorasan in northeastern Iran. The soil was completely homogeneous. Its geotechnical characteristics and classification are given in Table 1. XRD quantitative analysis was performed on the soil. Approximately 10 g of the soil was air-dried for 24 h and pulverized to pass through a No. 400 sieve (38 μm in diameter). The recognition of swelling clay minerals involved solubility in magnesium. Then, the soil was exposed to glycerol to enter the interlayer positions. It was heated at 550 °C for 1 h. The sample was then randomly placed on a glass slide and analyzed with the XRD test. The XRD test was conducted using the powder X-ray diffraction method, together with a PANalytical X'pert APD diffraction gauge (Philips pw3830 protractor) and a graphite monochromator. The device operated at a tube voltage of 40 kV, a current of 30 mA, and with a copper X-ray tube (λ = 1.5418 Å). Quantitative and qualitative analysis of XRD patterns was performed through the Philips Xpert Highscore plus (ICDD) version 3.0. software.

**Table 1.** Geotechnical characteristics and classification of bentonite soil.

| Property | Liquid Limit (%) | Plastic Limit (%) | Plasticity Index (%) | Sand (%) | Silt (%) | Clay (%) | United Soil Classification | Colour |
|---|---|---|---|---|---|---|---|---|
| | 396.2 | 40 | 356.2 | 0 | 22 | 78 | CH | White |

In addition to XRD, the microstructure of bentonite soil was examined using scanning electron microscopy (SEM). The bentonite clay particles were dried in an oven, powdered, and passed through a number 400 sieve. A silver-coated micrograph was obtained by a Jeol-Jsm 840A scanning electron microscope (JEOL Ltd., Tokyo, Japan).

To stabilize the bentonite soil, two additive types—cement as the chemical additive and epoxy resin as the binding additive of soil particles—were employed. The cement characteristics are given in Table 2. Epoxy resins are available in different viscosities and react with several curing agents or hardeners. The resin used in this research was RL440 with HY440 hardener from the Pars Composite Company located in Tehran. They are produced based on bisphenol A-type epoxy and a hardener of amine polymer. In the structure of this type of epoxy resin, a reactive diluent is used, which also improves the mechanical properties and impact resistance. Table 3 gives the physical and appearance characteristics of the epoxy resin.

**Table 2.** Chemical analysis of cement (%, by weight).

| Cement Type | $SiO_2$ | $Al_2O_3$ | $Fe_2O_3$ | CaO | MgO | $SO_3$ | Free CaO | Alkalies ($Na_2O\%$ + $0.658K_2O\%$) | L.O.I | Insoluble Residue |
|---|---|---|---|---|---|---|---|---|---|---|
| II | 21.50 | 4.95 | 3.97 | 63.52 | 1.75 | 2.20 | 1.4 | 1 | 1.19 | 0.5 |

**Table 3.** Physical and appearance characteristics of epoxy resin.

| Appearance | Color | The Combination Ratio of the Resin to the Hardener | Composition Time of the Components | Curing Time at Room Temperature | Density | Viscosity of the Compound at 25 °C |
|---|---|---|---|---|---|---|
| Liquid | Clear colorless | 2:1 | 5 min | 360 min | 1.1 $gr/cm^3$ | 440 cp |

### 2.2. Mix Design and Sample Preparation

Bentonite soil with different amounts of cement, including 4, 8 and 12% of the dry weight of the soil, was thoroughly mixed in the dry state until the mixture became uniform and homogeneous. In other words, the mentioned percentages of the primary soil were replaced by cement in practice. Based on the constructor's recommendation, the epoxy resin and hardener were blended at a 2:1 ratio and stirred with an electric mixer for 4 min to form a homogeneous, white-colored composition. The final composition obtained by mixing epoxy resin and hardener at the ratio of 2:1, which is a homogeneous and white-colored composition, is denoted by ER. In addition, distilled water is denoted by W in this study.

Epoxy resin and distilled water with different weight ratios of ER/W equal to 0, 1:4, 1:2, 1:1, and 2:1 were added to the homogeneous compound of bentonite soil and cement so that the total weight of both was equal to the optimum moisture weight. They were mixed according to the ASTM C938-97 standard [54] for 6 min with an electric mixer to achieve a homogeneous composition. The mixing was such that the epoxy resin was replaced by moisture. For example, in the mixture of bentonite soil, cement, and epoxy resin with the ER/W ratio equal to 0, the total weight amount of the optimum moisture was equal to that of distilled water. At the ER/W ratio of 2:1, the weight of the epoxy resin was twice that of distilled water, and the total weight of both was equal to the optimum moisture content. In addition, another mixing compound was made in which the total amount of optimum moisture was replaced by epoxy resin and added to bentonite soil not mixed with cement. Bentonite soil and epoxy resin were mixed similarly to the other compounds to obtain a homogeneous composition.

Before the compaction process, each compound was passed through a No. 10 sieve. In order not to affect the preparation method and sample compaction regarding the results of tests and also to examine the epoxy resin's impact on the compressibility of stabilized bentonite soil, the soil samples were compacted using the standard Proctor compaction conditions (constant energy equal to 0.055 $(kgm)/cm^3$). After compaction, each sample was wrapped in cellophane and placed in a plastic bag to prevent moisture loss. They were then cured at room temperature for 7 and 28 days.

### 2.3. Testing Methods and Required Parameters

The requirements of the ASTM D559 standard [55] were met as much as possible to expose the samples to successive W/D cycles. This standard is about examining the durability of soil–cement samples. Some experimental studies have adopted this standard to be compatible with their type of chemical stabilizer, soil, and the purpose of their research [56]. In the present study, after 7 and 28 days of curing, the stabilized samples were subjected to six cycles of W/D. Each cycle involved the soaking of stabilized samples for 24 h in water at room temperature before drying them for 24 h at 70 °C. In order to disintegrate the weak samples, they were exposed to a prolonged wetting period increased to 24 h compared to the ASTM D559 method, which set this period to 5 h.

The unconfined compressive strength (UCS) test based on ASTM D2166 [57] was carried out through a universal device model (ZWICK 1498). In this test, a load was applied to the soil sample at the strain rate of 1 mm/min. In addition to the high loading capacity, the device recorded the force values for deformation of 0.01 mm (equal to 0.00014 axial strain). The possibility of reporting small displacements led to the precise calculation of engineering behavior characteristics acquired from the stress–strain curve, including the stiffness and toughness of the material. Figure 1a–c show the universal device, how the load was applied, and the failure of the stabilized bentonite sample with 12% cement and epoxy resin with an ER/W ratio equal to 2:1 in the UCS test, respectively.

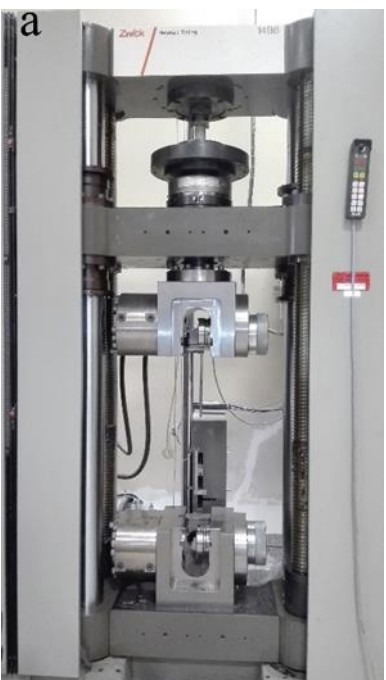
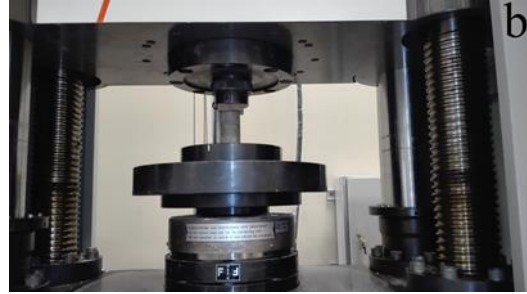
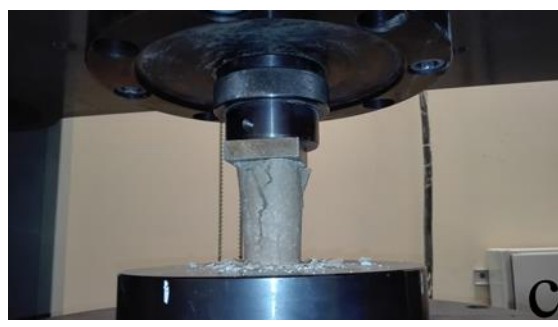

**Figure 1.** (**a**) Universal device; (**b**) how to apply the load; (**c**) failure of bentonite sample stabilized with 12% cement and epoxy resin with an ER/W ratio equal to 2:1.

In the UCS test, four parameters, namely, ultimate compressive strength, toughness, secant elastic modulus, and failure strain, were considered to evaluate the stress–strain behavior of the stabilized bentonite soil samples. The ultimate compressive strength is the maximum stress that material could withstand during the UCS test. It was denoted by $q_u$. The secant elastic modulus indicates the stiffness of the material. In this study, the strain calculated at 50% of the ultimate compressive strength was defined as the secant elastic modulus. It was indicated by $E_{50}$. The toughness is the strength of the material against failure. It is the energy that the matter can absorb. The surface area below the stress–strain curve indicates the soil energy absorption capacity, or the toughness of a material. The energy calculated up to the level of an arbitrary strain, e.g., up to the failure strain, does not distinguish between soils which are fundamentally different in behavior. Two soils may have approximately equal energy absorption capacity up to the failure point, but one has extremely brittle behavior, and the other has predominantly ductile behavior. Therefore, to make such a distinction, the energy absorption capacity along the entire stress–strain curve must be determined and compared. The fourth parameter was failure strain ($\varepsilon_f$), defined as the strain corresponding to the maximum uniaxial compressive strength. All the symbols used in this article are summarized in Table 4.

**Table 4.** Symbols and their descriptions used in this article.

| Symbol | Description | Symbol | Description |
|---|---|---|---|
| ER | The homogenous combination of epoxy resin and hardener with a ratio of 2:1 | $SP_{wetting}$ | The strength parameter of the stabilized bentonite sample with cement and epoxy resin after 24 h of wetting in the sixth cycle |
| W | Distilled water | $C_{SP}$ | $\frac{SP_{after\text{-}curing}}{SP_{wetting}}$ |
| $q_u$ | Ultimate compressive strength in the UCS test | $(SP)_e$ | $C_{SP} \times SP_{after\text{-}curing}$ |
| $\varepsilon_f$ | Failure strain | $(q_u)_e$ | The estimated ultimate compressive strength |
| $E_{50}$ | The secant elastic modulus at 50% of the ultimate compressive strength | $(E_{50})_e$ | The estimated secant elastic modulus |
| $SP_{after\text{-}curing}$ | The strength parameter of the stabilized bentonite sample with cement and epoxy resin after 7 days of curing | $(Toughness)_e$ | The estimated toughness |

## 3. Results and Discussion

### 3.1. Mineral Identification and Analysis of Bentonite Soil Using XRD and SEM Tests

Glycerol solubility transmits the 001-peak of magnesium-saturated smectite from 12–15 Angstroms to 17–18 Angstroms [58]. Heating the smectite to 300 °C and higher temperatures leads to water evaporation and transfers the 001-peak to 10 Angstroms [58]. According to Figure 2a,b, the solubility of glycerol changed the 001-peak of magnesium-saturated bentonite soil from 12.88 Angstroms to 17.73 Angstroms. Then, heating to 550 °C changed the 001-peak to 9.89 Angstroms. Therefore, the predominant clay mineral of the bentonite soil was montmorillonite. The quantitative analysis of X-ray diffraction patterns carried out on the bentonite soil is provided in Table 5.

**Table 5.** Mineralogical composition of bentonite soil (wt% of the total amount).

| Montmorillonite (%) | Quartz (%) | Albite (%) | Cristobalite (%) |
|---|---|---|---|
| $Ca_{0.2}(Al,Mg)_2Si_4O_{10}(OH)_2,xH_2O$ | $SiO_2$ | $(Na,Ca)(Si,Al)_4O_8$ | $SiO_2$ |
| 45 | 14 | 26 | 12 |

Figure 2c is an SEM image of bentonite clay showing the mineral texture of montmorillonite. The SEM image was taken at the magnification capacity of 30KX (scale bar = 1 μm). It shows the turbulent surface texture as masses of wavy, filmy particles [59,60]. While the boundaries of the raised particles above the overall turbulent surface are entirely sharp, it is not easy to identify the boundaries of individual particles mixed into the mass. Montmorillonite particles are often described and plotted as very lean plates. It is clear that the actual particles in montmorillonite are skinny, but they are not in the form of the plates [61].

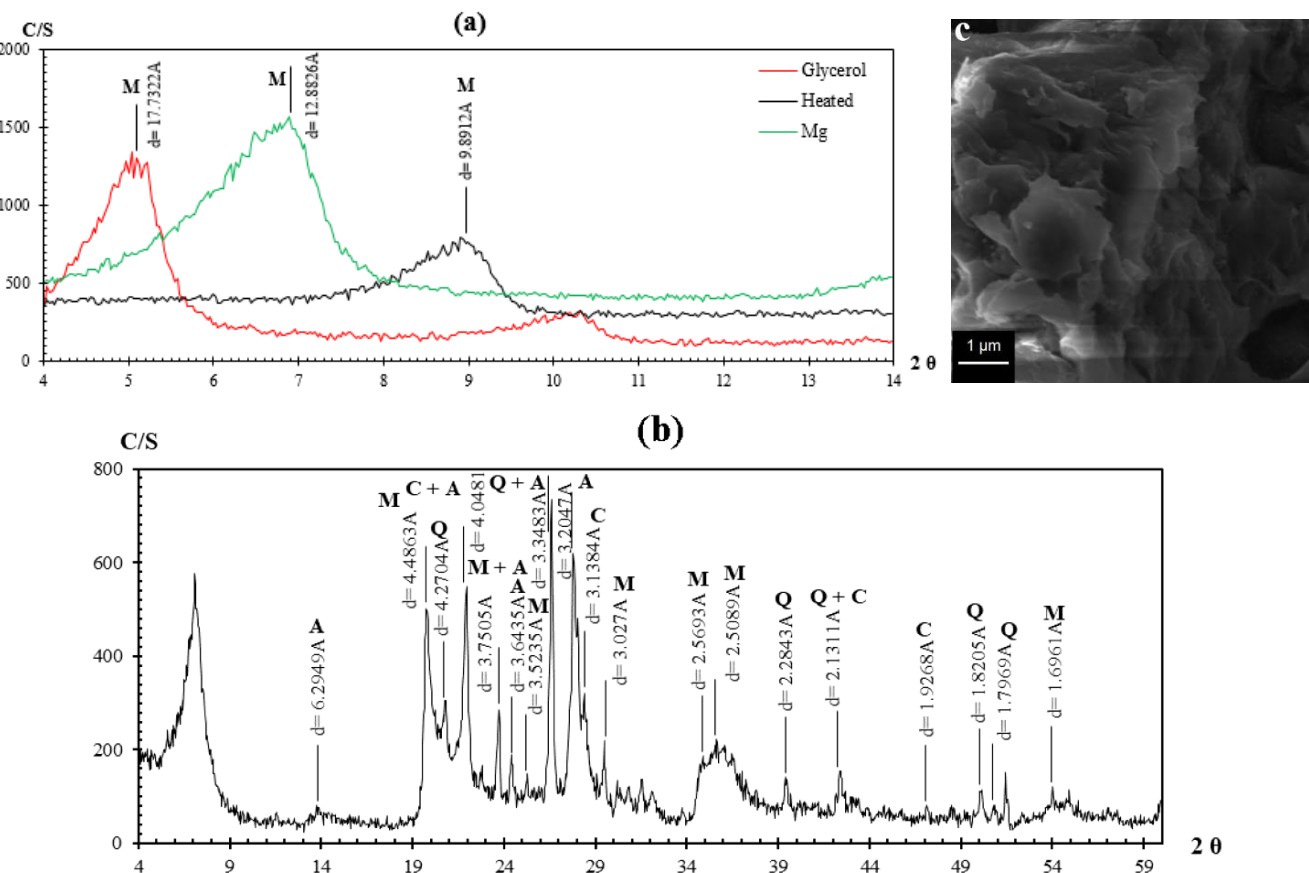

**Figure 2.** (**a**) X-ray diffraction of Mg-solvation, glycerol-solvation, and heated clay fractions of the bentonite soil; (**b**) air-dried bentonite soil containing montmorillonite (M), cristobalite (C), quartz (Q), and albite (A), d-spacing in Angstroms (d); (**c**) SEM images of bentonite soil with magnification at 30 KX.

*3.2. Cement Stabilization*

3.2.1. Strength Parameter Evaluation

For each stabilized bentonite sample, three control samples were made and subjected to UCS testing. The toughness, $q_u$, $\varepsilon_f$, and $E_{50}$ values are considered the mean values of the accepted tests. The error bars in some of the diagrams of this research show the standard deviation error. They show the variability of the data of the three tests performed for each stabilized bentonite sample. Thus, the error bar shows how close the data of the three tests for each stabilized sample are to the average content.

Initially, the bentonite soil was stabilized with 4, 8, and 12% cement and cured for 7 and 28 days. The results of the UCS tests conducted on the samples are shown in Figure 3. Figure 3a shows their stress–strain curves. Diagrams of changes in their amounts of $q_u$, toughness, $\varepsilon_f$, and $E_{50}$ are displayed in Figure 3b–e, respectively. The bentonite clay soil sample had significant ductility due to its high water absorption and completely plastic behavior. It showed failure strain with a value of about 5%. The applied stress after the failure point caused it to take on the shape of a barrel, and no crack was created on it. The unstabilized bentonite sample was not evaluated as suitable due to its low strength and high strain. With the addition of 4% cement and curing for 7 days, the $E_{50}$ and $q_u$ values of the bentonite soil increased by 5 and 6 times, respectively, so that the toughness value against failure tripled. After 28 days of curing, the $q_u$, $\varepsilon_f$, and $E_{50}$ values compared to 7 days of curing increased by 65, 22, and 72%, respectively. As a result, the toughness value increased by 51%.

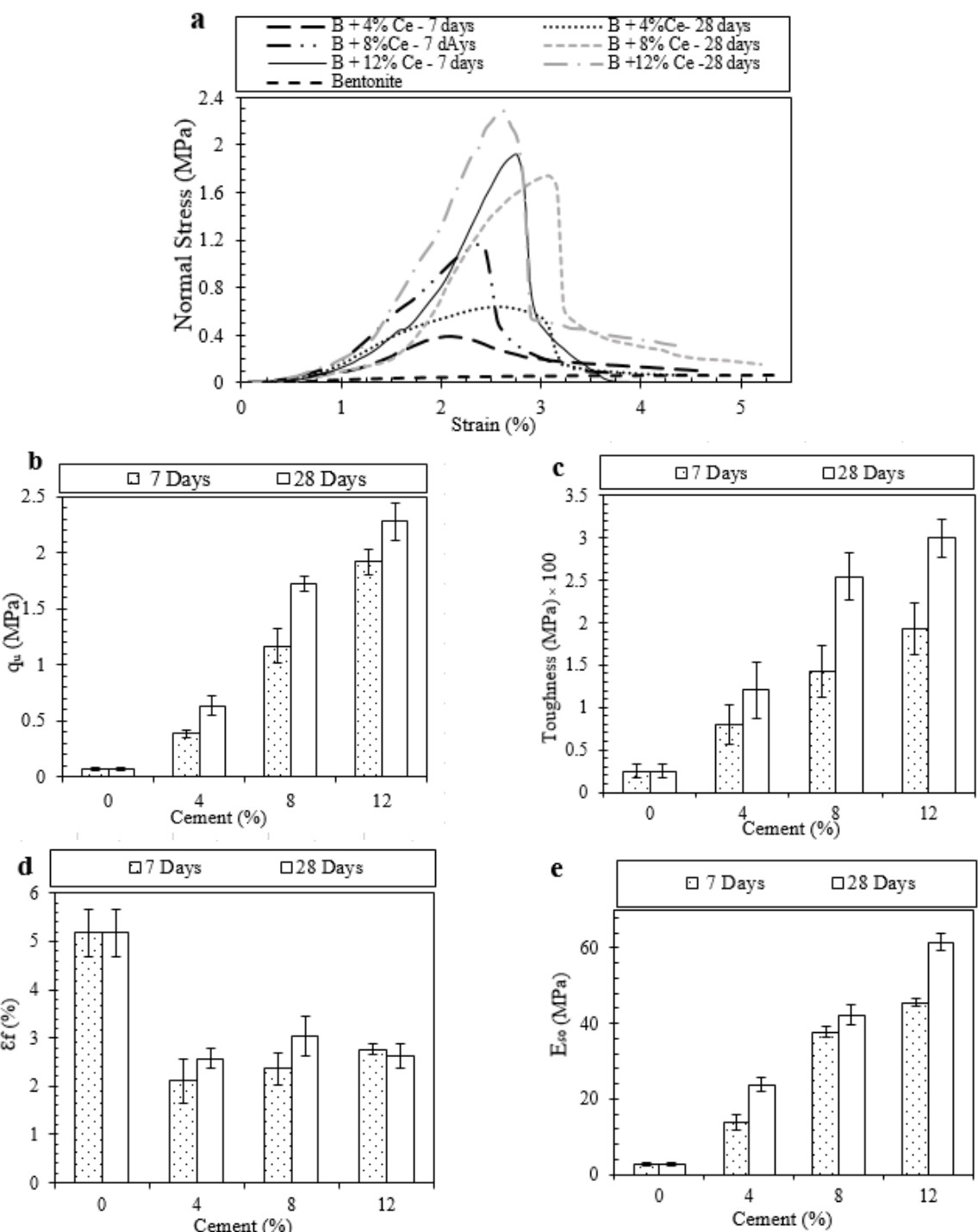

**Figure 3.** (**a**) The stress–strain curves; the changes of (**b**) qu; (**c**) toughness; (**d**) failure strain ($\varepsilon_f$); and (**e**) secant elastic modulus (E50) values of the stabilized bentonite soil for different percentages of cement after 7 and 28 days of curing.

By adding 8% cement and curing the sample for 7 days, the $q_u$ and $E_{50}$ values became 3 and 2.7 times those of the bentonite sample with 4% cement, respectively. Its stress–strain curve was on top of that of the 7-day stabilized sample with 4% cement. As a result, the amount of toughness increased by 80%. For the stabilized sample with 8% cement after 28 days of curing, the $q_u$, $E_{50}$, and toughness values increased by 48, 12, and 78% compared to the 7-day stabilized sample, respectively. Considering that the increases of the $q_u$ and stiffness values were less than 50%, this significant increase of the toughness value of the

28-day stabilized sample compared to the 7-day stabilized sample was due to the rise of the $\varepsilon_f$ value by 27.8%. After 7 days of curing, the $q_u$, $E_{50}$, and toughness values of the stabilized sample with 12% cement compared to the stabilized sample with 8% cement increased by 65, 21, and 35%, respectively. They increased by 32, 46, and 18% after 28 days of curing, respectively. These increases of the $q_u$, $E_{50}$, and toughness values of the sample with 12% cement compared to the sample with 8% cement against the increases of these parameters for the sample with 8% cement compared to the sample with 4% cement were negligible. So, the optimum cement content was 8%, which was economical. By adding more cement than 4%, the $q_u$ and also $\varepsilon_f$ values increased, but the increase of the $\varepsilon_f$ value was not significant. The stabilization of natural bentonite with cement caused the failure strain values of both 7-day and 28-day stabilized samples to decrease by about 40% and reach less than 3%. According to their stress–strain curves in Figure 3a, the stress dropped sharply after the failure point. Despite the ductile behavior of the natural bentonite sample, the cement stabilization resulted in brittle behavior.

The curing time had little effect on the strength parameter values of the sample with 8% cement. The $q_u$ and $E_{50}$ values of the 28-day stabilized sample with 12% cement increased by 19 and 35%, respectively, compared to the 7-day stabilized sample, which were slight amounts. After 28 days of curing, the increase of the strength parameters values for both stabilized samples with 8 and 12% cement was less than 47% compared to the corresponding values of their 7-day stabilized samples. So, the pozzolanic reactions that increase the strength and stiffness values to more than 70% in the long term had not occurred or had occurred at a weak level. The long-time mechanical and engineering properties of chemically stabilized soils depend on the complex reactions of soil mineralogy and chemical additives along with environmental effects [62–64]. Large amounts of montmorillonite induce less effectiveness of the chemical stabilization of soils because this mineral prevents the pozzolanic reactions between the soil and the stabilizer [65]. In addition, their existence creates more hydration among the clay matrixes, resulting in the loss of cementation skeleton and reactions [66]. Considering the plasticity properties of clay soils regardless of their mineralogy leads to poor performance of chemical stabilization of soils rich in montmorillonite [67]. According to Table 5, the bentonite soil contained approximately 45 wt% montmorillonite mineral out of the total amount.

Therefore, the results of the mentioned studies confirmed the results of the UCS tests, and based on both of them, the high content of clay minerals not only prevented the complete mixing of the cement with the soil, but it even increased the cement content needed to stabilize. In general, if the range of the soil PI is more than 30%, the mixing of soil with cement is difficult, and stabilization is not adequate [68]. After 28 days of curing, the sample with 12% cement showed minimal progress of pozzolanic reactions. Therefore, by increasing the cement percentage, the curing time slightly affected pozzolanic reactions between the cement and the bentonite soil containing the predominant mineral of montmorillonite. The samples gained their maximum strength parameters within 7 days of curing.

### 3.2.2. Evaluation of Durability against W/D Cycles

The cement-stabilized bentonite samples were subjected to the durability process after 28 days of curing. As soon as the stabilized samples with 4 and 8% cement were soaked in water in the first cycle, they collapsed. The surface of the stabilized sample with 12% cement was filled with cracks due to the severe shrinkage of the bentonite soil after 24 h of drying in the first cycle. These cracks were not superficial and were deep. As soon as the sample was immersed in water for the second cycle, it collapsed (Figure 4a,b). In order to find the cement percentage with which bentonite soil was stabilized to last up to 6 cycles, the 28-day stabilized samples with 20 and 30% cement were also tested. Still, they only lasted up to two cycles of W/D. The pozzolanic reactions that occur over time by exchanging cations between the soil particles and cement lead to cement compound formation. These compounds increase the strength parameter values of the 28-day stabilized samples compared

to the 7-day stabilized samples by more than 70%. Also, they bind the soil particles to each other to provide strength and rigidity to the stabilized sample, allowing it to last against W/D cycles. The pozzolanic reactions in the stabilization of bentonite soil with cement had not been performed, or they had been performed at a poor level. This was due to the slight increase in the long-term strength parameters of the cement-stabilized bentonite soil samples compared to their short-termb strength parameters. On the other hand, the 28-day stabilized samples did not withstand 6 cycles of W/D even with 30% cement.

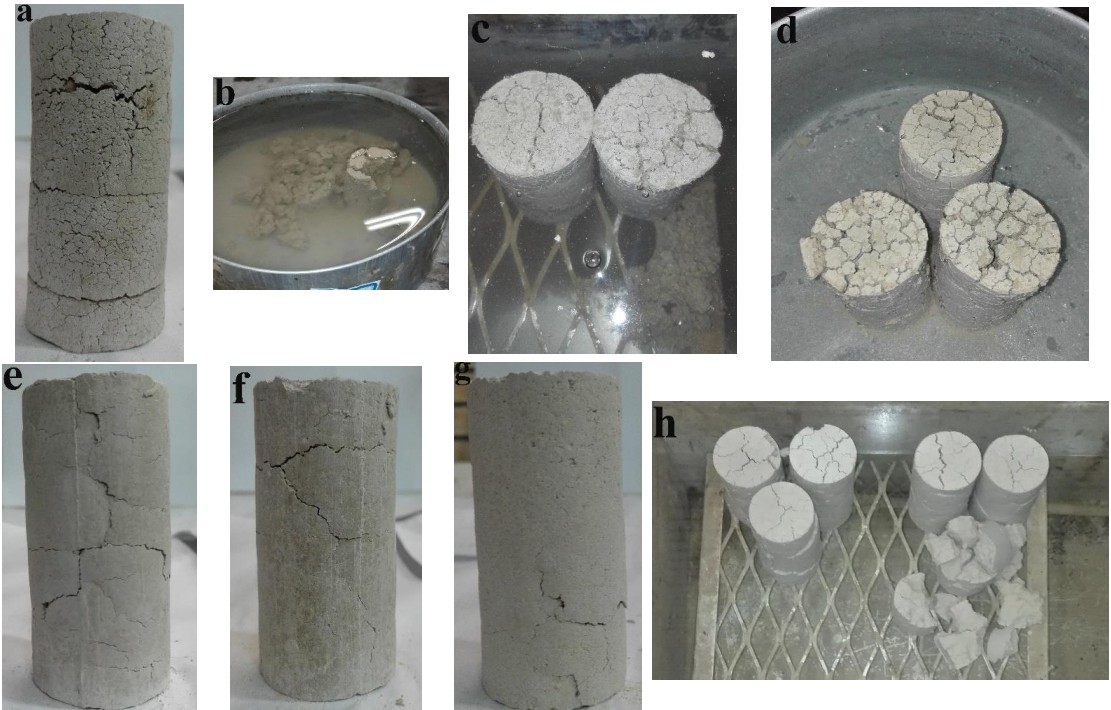

**Figure 4.** (**a**) B + 12% Ce 28-day sample after 24 h of drying in the first cycle; (**b**) B + 12% Ce 28-day sample disintegrated as soon as it was soaked in the water for the second cycle; (**c**) B + 8% Ce, ER/W = 1:4 sample at the wetting phase in the second cycle; (**d**) B + 4% Ce, ER/W = 1:2 sample at the wetting phase of the second cycle; (**e**) B + 12% Ce, ER/W = 1:4 sample after 24 h of drying in the second cycle; (**f**) B + 8% Ce, ER/W = 1:2 sample after 24 h of drying in the second cycle; (**g**) B + 12% Ce, ER/W = 1:2 sample after 24 h of drying in the second cycle; (**h**) at an ER/W ratio of 0.25, the sample with 12% cement and at an ER/W ratio of 0.5, the samples with 8 and 12% cement decomposed after 24 h of soaking in the third cycle; B: bentonite soil.

Therefore, the results of durability tests of entonite clay samples stabilized with cement in the present study confirmed the low amount of pozzolanic reaction of bentonite soil with cement. This had been confirmed by previous studies [49,69–71]. According to these studies, for smectite clays such as montmorillonite, pozzolanic reaction with cement occurred at a very low level.

### 3.3. Stabilization with Cement and Epoxy Resin

In order to find the lowest amount of epoxy resin that led to the cement-stabilized samples not collapsing after six cycles of W/D, the addition of epoxy resin was started from an ER/W ratio equal to 1:4. The cement-stabilized samples were then tested by adding epoxy resin with ER/W ratios of 1:2, 1:1, and 2:1. The results of the uniaxial tests performed on 7-day cement- and epoxy resin-stabilized samples with ER/W ratios of 1:4, 1:2, 1:1, and 2:1 are shown in Figure 5. Figure 5c,d,f show the changes in the values of $q_u$, toughness, $\varepsilon_f$, and $E_{50}$ of the stabilized samples with 4, 8, and 12% cement according to different ratios of the ER/W, respectively.

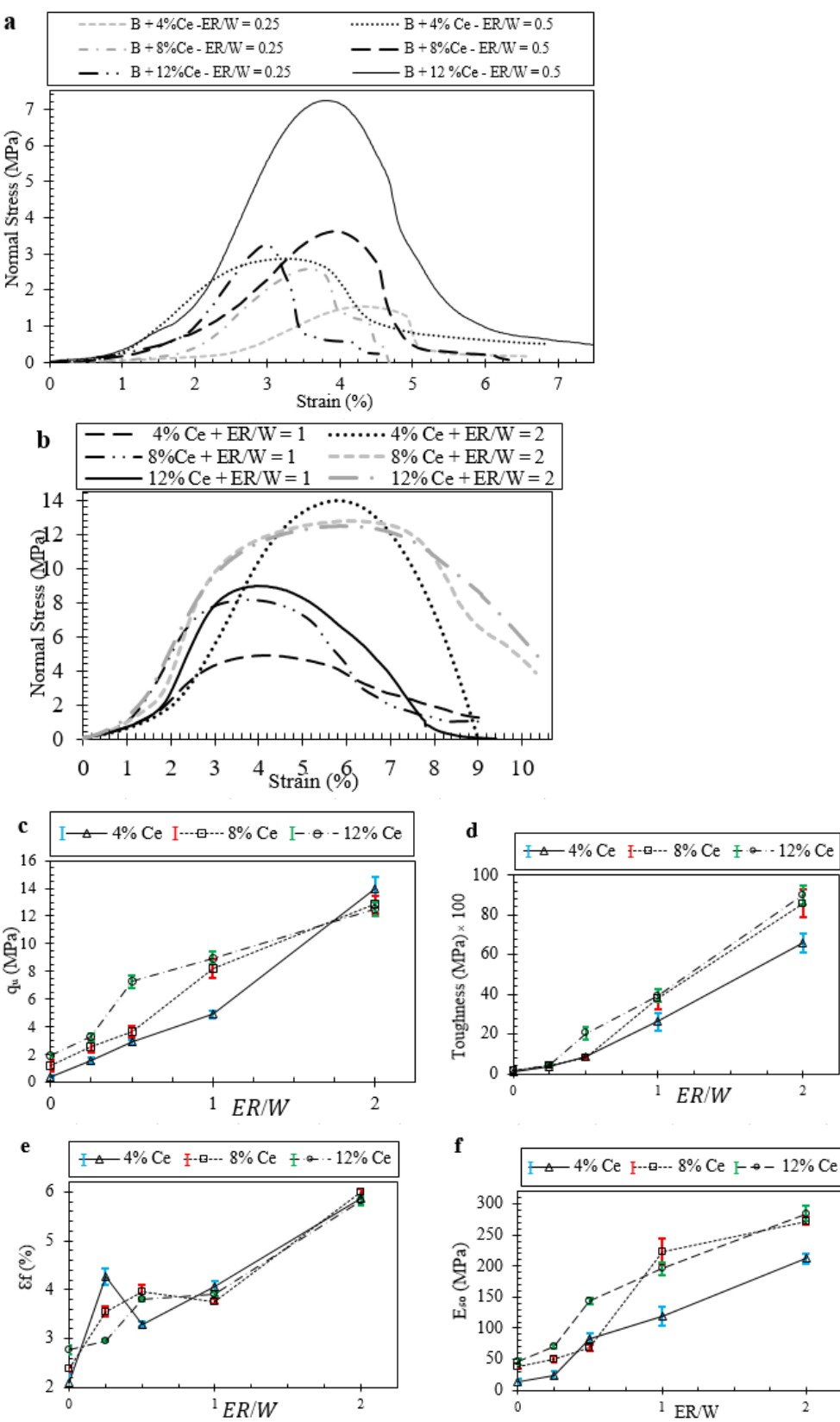

**Figure 5.** The stress–strain curves at the ER/W ratios equal to (**a**) 1:4 and 1:2; (**b**) 1:1 and 1:2; The changes of (**c**) $q_u$; (**d**) toughness; (**e**) failure strain ($\varepsilon_f$); and (**f**) secant elastic modulus ($E_{50}$) values of the stabilized bentonite soil with different percentages of cement and according to different ratios of ER/W.

### 3.3.1. Investigation of the Strength Parameters after 7 Days of Curing

- Cement and epoxy resin with an ER/W ratio equal to 1:4

According to the stress–strain curves in Figure 5a, unlike the 7-day cement-stabilized samples without epoxy resin, the stress of cement- and epoxy resin-stabilized samples with an ER/W ratio equal to 1:4 did not experience a sudden, sharp drop after the failure point. The addition of epoxy resin with an ER/W ratio equal to 1:4 prevented the sudden, sharp decline of the stress after the failure point in the stress–strain curves. So, the samples' toughness against failure increased. At an ER/W ratio of 1:4 and for different percentages of cement, the amount of failure strain varied from 3 to 5%, while for 7-day cement-stabilized samples without epoxy resin, it was less than 3%. As a result, the failure of cement- and epoxy resin-stabilized bentonite soil samples with an ER/W ratio of 1:4 was much more ductile compared to the cement-stabilized samples without epoxy resin.

For the stabilized samples with 4, 8, and 12% cement at an ER/W ratio equal to 1:4, the increases of $q_u$, $E_{50}$, and toughness were 68% to 4 times, 2.3 to 4.1 times, and 32 to 74%, respectively, compared to 7-day cement-stabilized samples without epoxy resin. The presence of epoxy resin at an ER/W ratio equal to 1:4 in cement- and epoxy resin-stabilized bentonite samples increased the strength and toughness values of them to several times those of the 7-day cement-stabilized samples without epoxy resin. By adding 8% cement, the $q_u$, toughness, and $E_{50}$ values increased by 66%, 30%, and 2 times, respectively, compared to the sample with 4% cement, while by adding 4% more cement (i.e., stabilization with 12% cement), these corresponding values increased by 25, 4 and 42%, respectively. Therefore, the optimum amount of cement when adding epoxy resin with tan ER/W ratio equal to 1:4 was 8%.

With the addition of cement, the $\varepsilon_f$ values decreased so that by adding 12% cement, they decreased by 30% compared to adding 4% cement. So, at an ER/W ratio of 1:4, the failure values of the samples stabilized by adding cement were brittle compared to each other. The toughness values against failure by adding 8 and 12% cement were in the range of the stabilized sample with 4% cement. This was due to the high value of the $\varepsilon_f$ of the sample with 4% cement, and it was not evaluated as suitable in terms of strength.

- Cement and epoxy resin with an ER/W ratio equal to 1:2

According to the stress–strain curves shown in Figure 5a, the stress after the failure point for the cement-stabilized samples with an ER/W ratio of 1:2 experienced a much smaller sudden drop compared to the stabilized samples with similar cement percentages and with an ER/W ratio of 1:4. The smaller sudden drop indicated that the failure of the cement-stabilized samples with an ER/W ratio of 1:2 was more ductile than in the stabilized samples with an ER/W ratio of 1:4. In addition, by adding epoxy resin at an ER/W ratio of 1:4 to 1:2 and for different percentages of cement, the values of $q_u$, $E_{50}$, and toughness increased in the range of 85% to 2 times, 36% to 3.4 times, and 80% to 4.6 times, respectively.

With the addition of cement from 4 to 8% at an ER/W ratio of 1:2, the value of $q_u$ increased by less than 27%, and the values of toughness and $E_{50}$ not only did not increase but even decreased by less than 20%. With the addition of 8% cement, the strength parameter values had slight changes, but with 12% cement, they increased significantly. The toughness and $E_{50}$ values of the sample with 12% cement became 2.6 and 2.1 times those of the sample with 8% cement, respectively. Therefore, at an ER/W ratio equal to 1:2, the optimum content of cement was 12%. At an ER/W ratio of 1:2, unlike an ER/W ratio of 1:4, the failure strain value increased with the increase of the cement percentage, and it reached a maximum value of 4%.

- Cement and epoxy resin with an ER/W ratio equal to 1:1

According to the stress–strain curves shown in Figure 5b, the stress after the failure point did not decrease abruptly for the cement- and epoxy resin-stabilized samples with an ER/W ratio equal to 1:1. Its reduction rate considerably decreased so that at the strain interval with a length of about 1% after the failure point, the stress drop rate was

almost zero. A plastic region expanded at the strain interval with a length of about 2%, in which the difference between the sample stress and the $q_u$ value was negligible. As a result, the strength of the samples against failure was considerable, and they had failures with significant ductility. By increasing the ER/W ratio from 1:2 to 1:1 and for different percentages of cement, the $q_u$, $E_{50}$, and toughness values increased in the range of 70% to 2.25 times, 37% to 3.3 times, and 93% to 5 times, respectively. The changes that were more evident by increasing the epoxy resin from an ER/W ratio of 1:2 to 1:1 were the significant increase of the ductility and toughness. At the same time, the strength parameters also had high values.

According to the stress–strain curves in Figure 5b, at an ER/W ratio of 1:1, the first part of the stress–strain curve until the failure point for the sample with 12% cement was above that of the sample with 4% cement, and for the sample with 8% cement, it was on top of that of the sample with 12% cement. Therefore, the stress increase rate of the sample with 8% cement until the failure point was significantly higher than those of the samples with 4 and 12% cement. The stress–strain curve of the sample with 8% cement from the strain of 0 to about 6% was on top of that of the sample with 4% cement by a significant distance. In addition, its stress–strain curve differed slightly from that of the sample with 12% cement at many strain intervals. According to the stress–strain curves, at an ER/W ratio equal to 1:1, it was expected that the optimum content of cement would be 8%.

Although by adding 8% cement at an ER/W ratio equal to 1:1, the $q_u$, toughness, and $E_{50}$ values increased by 68, 44, and 88%, respectively, compared to the sample with 4% cement, with the addition of 4% more cement (i.e., stabilization with 12% cement), the $q_u$ and toughness values increased by 9 and 4.5%, respectively. Even the hardness value decreased by 12.2%. Therefore, by adding 12% cement, the rates of increase in the $q_u$, toughness, and hardness values were stopped, and the optimum amount of cement was 8%. With the addition of 8% cement at an ER/W ratio of 1:1, the $\varepsilon_f$ amount decreased by less than 8%, and by adding 12% cement, the $\varepsilon_f$ value change was less than 4%. Therefore, at an ER/W ratio equal to 1:1, increasing the cement content by more than 4% had a negligible effect on the failure strain value, and so it remained almost constant. For different percentages of cement, it was almost less than 4%.

- Cement and epoxy resin with an ER/W ratio equal to 2:1

By increasing the epoxy resin content from an ER/W ratio of 1:1 to 2:1 and for different percentages of cement, the value of $q_u$ increased in the range of 40% to 2.85 times, and the stiffness value increased in the range of 22 to 78%. The change that was more evident by increasing the ER/W ratio to 2:1 was the increase in the failure strain value in the range of 44 to 60%. Consequently, the increase in the toughness value was in the range of 2.3 to 2.5 times for different cement percentages. As a result, the cement- and epoxy resin-stabilized bentonite samples with an ER/W ratio of 2:1 had high strength. At the same time, they had significant ductility and high toughness against failure.

In two studies conducted by Hamidi and Marandi [23,24] and Ghiyas and Bagheripour [51], the strength properties of two stabilized clays with cement and epoxy resin, including kaolin and bentonite, were investigated. Both studies concluded that for the kaolin clay sample, ductility increased more significantly than strength, while for the bentonite sample, the strength and ductility increased significantly and simultaneously. Therefore, the results of the present study were consistent with the results of the studies conducted by Hamidi and Marandi [23,24] and Ghiyas and Bagheripour [51] and confirmed that bentonite soil with a high plasticity limit in combination with cement and epoxy resin works effectively both in terms of strength and ductility.

According to the stress–strain curves of Figure 5b, at an ER/W ratio equal to 2:1, the stress–strain curve of the sample with 8% cement at the strain interval from 0 to approximately 4% was on top of that of the sample with 4% cement by a considerable distance. The stress–strain curve of the sample with 12% cement almost, with a slight difference, matched the stress–strain curve of the sample with 8% cement. The stress after the failure point for the sample with 4% cement decreased sharply compared to the

samples with 8 and 12% cement. However, the stress in the vicinity of the failure point for the samples with 8 and 12% cement at the strain interval with a length of approximately 3% was in the range of a $q_u$ value equal to 12 MPa. As a result, significant broadness of the plastic region was achieved at the strain interval with a length of approximately 3%. Therefore, by adding epoxy resin such that the ER/W ratio was equal to 2:1, the ductility and strength of samples with 8 and 12% cement increased significantly.

By adding 8% cement at an ER/W ratio equal to 2:1, the $q_u$ value decreased slightly by 8% compared to the sample with 4% cement. The toughness and hardness values increased by 30 and 28%, respectively. Although the increase in the strength parameter values of the sample with 8% cement compared to the sample with 4% cement was less than 31%, its stress at the large strain interval was close to the $q_u$ value, and its drop rate after the failure point was low. This was indicative of the high ductility and strength of the sample with 8% cement against failure. Therefore, at an ER/W ratio equal to 2:1, the optimum amount of cement was 8%.

3.3.2. Evaluation of Durability against W/D Cycles

- Cement and epoxy resin with ER/W ratios of 1:4 and 1:2

Deep cracks were created on the surfaces of the samples stabilized with 4 and 8% cement and epoxy resin with an ER/W ratio equal to 1:4 due to the high shrinkage of bentonite soil during drying in the first cycle. As shown in Figure 4c, the sample with 8% cement lost rigidity in the wetting phase of the second cycle. When the water penetrated through deep cracks into the sample, it gradually disintegrated.

After a few hours, it disintegrated completely. Despite the significant increase of the strength parameter values of the samples with 4 and 8% cement due to adding epoxy resin with an ER/W ratio equal to 1:4, they did not last longer than one cycle. They collapsed in the wetting phase of the second cycle.

At an ER/W ratio equal to 1:4, the optimum amount of cement was 8%. Adding 4% more cement (i.e., stabilization with 12% cement) did not increase the after-curing strength parameters of the sample but improved the sample durability. It lasted through the second cycle of W/D. The shrinkage potential of the bentonite soil was very high, and many deep cracks appeared on the sample's surface after 24 h of drying in the second cycle (Figure 4e). As shown in Figure 4h, the sample collapsed as soon as it was immersed in water for the third cycle. After increasing the epoxy resin from an ER/W ratio of 1:4 to 1:2, the sample with 4% cement still did not last. As shown in Figure 4d, it disintegrated in the wetting phase of the second cycle. Although the after-curing strength parameters of the sample with 8% cement were slightly different from those of the sample with 4% cement, it lasted through the second cycle of W/D. Therefore, increasing the percentage of cement at an ER/W ratio equal to 1:2 improved the durability of the samples.

The optimum amount of cement at an ER/W ratio equal to 1:2 was 12%. This sample lasted until the second cycle of W/D. By comparing its image after 24 h of drying in the second cycle in Figure 4g with the sample with 8% cement in Figure 4f, it was observed that there were fewer cracks on its surface. Still, like the sample with 8% cement, as shown in Figure 4h, it disintegrated after a few hours in the wetting phase of the third cycle. The main point to consider in this part is that the $q_u$ value of the sample reached approximately 7 Mpa. Despite the considerable $q_u$ value, it did not overcome the swelling–shrinkage potential, nor did it retain its stiffness during the wetting phase in the third cycle, and it collapsed.

According to Figure 4, the development of deep cracks on the surface of the cement- and epoxy resin-stabilized samples with ER/W ratios of 1:4 and 1:2 was due to the severe shrinkage of the bentonite soil during drying in the early cycles. As soon as the water penetrated these samples through the cracks in the wetting phase of the primary cycles, they lost their cohesiveness and collapsed. Upon increasing the ER/W ratio to the values of 1:1 and 2:1 for different percentages of cement, despite the severe shrinkage of the bentonite soil, the stabilized sample retained its rigidity and did not collapse. The cementing and

bonding materials produced by the stabilization held the bentonite soil particles and stuck them together. The better the stabilization, the more substantial the cementitious and adhesive materials that were made. So, fewer cracks on the sample surface were created during the drying phase of each cycle, and less water penetrated the sample during the wetting cycle. When water penetrated the stabilized sample, the moisture came into contact with the bentonite clay, and so it became softer, resulting in high strain and low strength.

The cement- and epoxy resin-stabilized samples with ER/W ratios of 1:1 and 2:1 achieved the durability standard of this study. They lasted for up to six cycles of W/D. In order to evaluate the effect of W/D cycles on the strength parameters of these stabilized samples, uniaxial tests were performed on them after 24 h of wetting in the third and sixth cycles. The results of their uniaxial tests, including the stress–strain curves and the changes of the strength parameter values at the end of wetting in the third and sixth cycles, are shown in Figures 6 and 7, respectively.

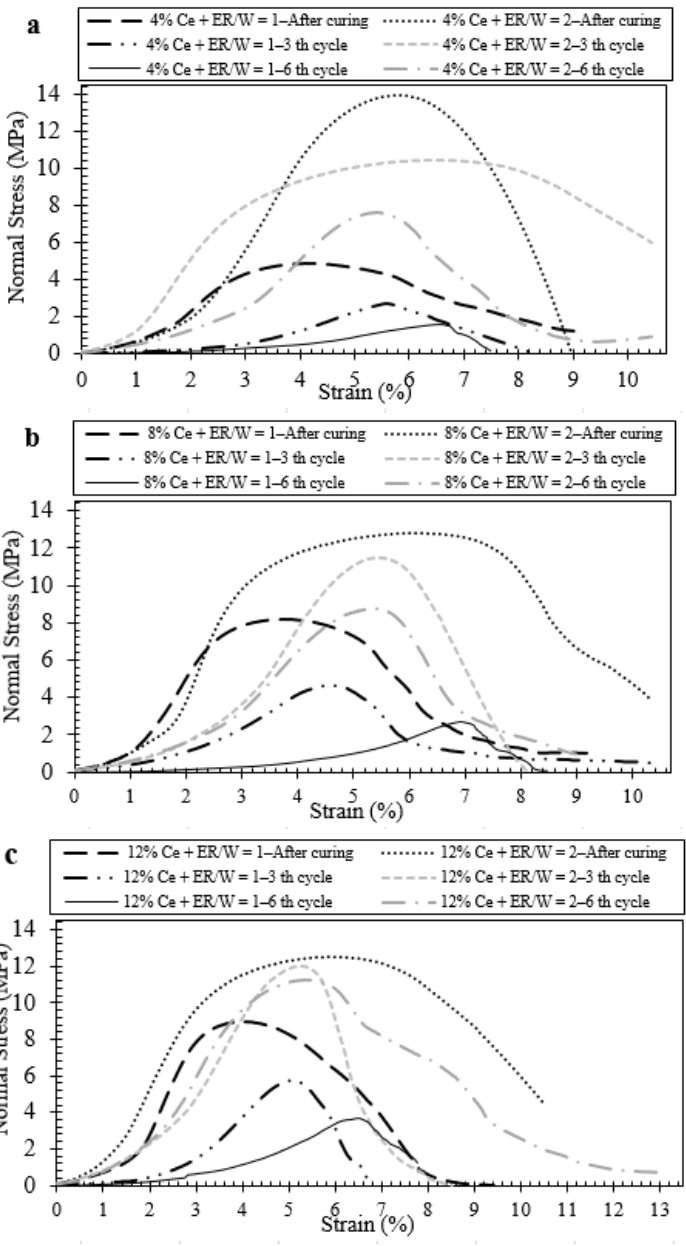

**Figure 6.** The stress–strain curves of stabilized bentonite samples with ER/W ratios equal to 1:1 and 2:1 and (**a**) 4% cement; (**b**) 8% cement; and (**c**) 12% cement at the end of wetting in the third and sixth cycles.

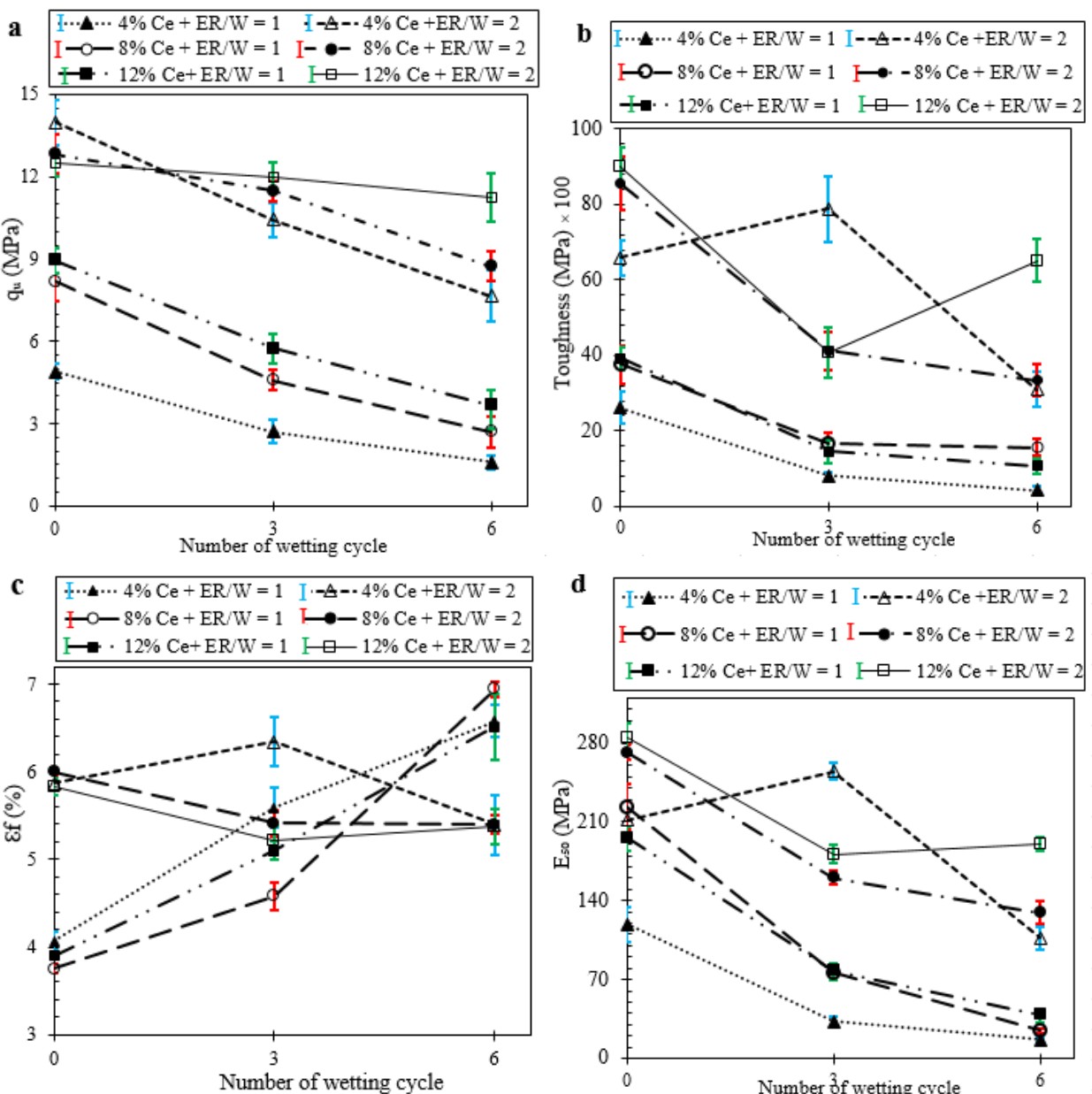

**Figure 7.** The changes in the (**a**) $q_u$; (**b**) toughness; (**c**) failure strain ($\varepsilon_f$); and (**d**) secant elastic modulus ($E_{50}$) values of the stabilized bentonite samples with different percentages of cement and epoxy resin at ER/W ratios equal to 1 and 2 after 24 h of soaking in the third and sixth cycles.

- Indirect estimation of the swelling–shrinkage potential of bentonite soil

In order to estimate the swelling–shrinkage potential of bentonite soil, the durability results of the stabilized bentonite samples that lasted for up to six cycles of W/D were employed. The strength parameter values required to overcome the swelling–shrinkage potential of bentonite soil were estimated using the following equations:

$$C_{SP} = \frac{SP_{\text{after-curing}}}{SP_{\text{wetting}}} \tag{1}$$

$$(SP)_e = C_{SP} \times SP_{\text{after-curing}} \tag{2}$$

where $SP_{\text{after-curing}}$ is the value of the strength parameter of the stabilized bentonite sample with cement and epoxy resin after 7 days of curing, and $SP_{\text{wetting}}$ is its corresponding value

after 24 h of wetting in the sixth cycle. $C_{SP}$ is the estimated coefficient of the strength parameter calculated to obtain $(SP)_e$. The strength parameters were $(q_u)_e$, $(E_{50})_e$, and $(toughness)_e$, estimated using Equation (2).

After stabilization of the bentonite soil by any method such as the combined use of chemical and adhesive additives, it was predicted that the strength parameters had to attain at least the values of $(q_u)_e$, $(E_{50})_e$, and $(toughness)_e$ to overcome the swelling–shrinkage potential of the bentonite soil. The overcoming of the swelling–shrinkage potential of the bentonite soil to a level such that no cracks were created on the surface of the bentonite soil during drying in each cycle was the target. The water did not penetrate the sample through these cracks during wetting, nor did it become as soft and low-strength as the wet clay.

- Cement and epoxy resin with an ER/W ratio equal to 1:1

According to Figure 6a, the stress–strain curve of the bentonite sample stabilized with 4% cement and epoxy resin with an ER/W ratio of 1:1 significantly dropped at the end of wetting in the third cycle. Also, the stress–strain curve of the sample after six cycles of W/D was at a considerable distance below that after three cycles of W/D. According to Figure 7, the $q_u$ and $E_{50}$ values of this sample after three cycles decreased by 44.8 and 72.7%, and after six cycles, they decreased by 67.5 and 86%, respectively. In addition, after three cycles, the $\varepsilon_f$ value of this sample increased by 37.4%, and the toughness value decreased by 68.8%. After six cycles, the $\varepsilon_f$ value increased by 62%, and the toughness value decreased by 84%.

The development of cracks on the sample surface was low and was not visible until the drying phase of the second cycle. From the third cycle of drying, the cracks became highly visible. The images of the bentonite sample stabilized with 4% cement and an ER/W ratio of 1:1, from the end of drying in the third cycle to the end of the wetting in the sixth cycle, are shown in Figure 8. Although the crack development, as shown in Figure 8c, appeared to divide the sample into two parts, it was on the shell of the sample surface, and it was not deep, which wouldd have caused the sample to collapse and lose rigidity. After 24 h of submerging the sample in the fifth cycle, more water had penetrated the sample through the cracks so that the width of the cracks increased after 24 h of drying of the sample, as shown in Figure 8e. This was due to the severe shrinkage of the bentonite soil and the loss of more water that penetrated the sample in the previous cycle. These cracks were on the surface shell of the sample. With the sequence of cycles, their depth on the sample surface increased but did not cause it to lose rigidity and collapse.

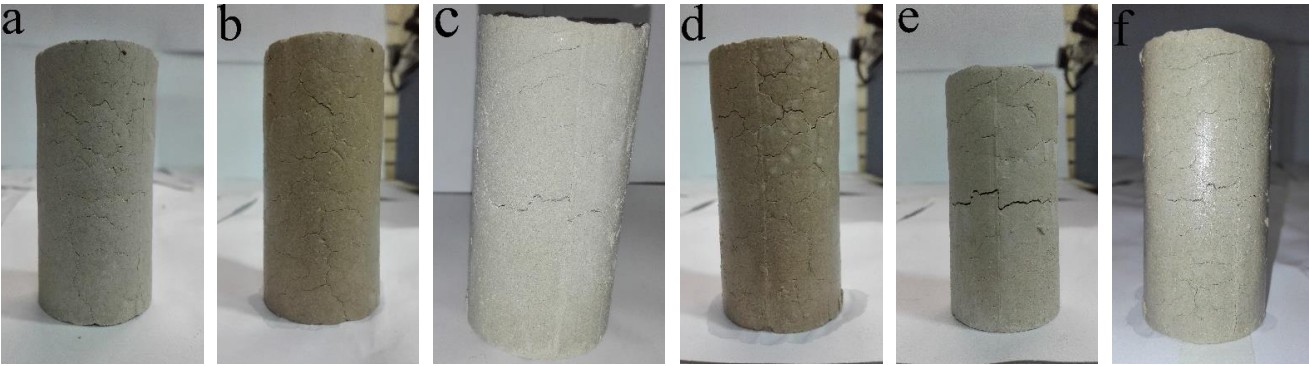

**Figure 8.** The stabilized bentonite sample with 4% cement and epoxy resin with an ER/W ratio equal to 1 at the end of (**a**) drying of the third cycle; (**b**) wetting of the fourth cycle; (**c**) drying of the fourth cycle; (**d**) wetting of the fifth cycle; (**e**) drying of the fifth cycle; and (**f**) wetting of the sixth cycle.

By applying more cycles, the strength and rigidity of the sample with 4% cement decreased due to cracks created in the drying phase of each cycle and water penetration into the sample through these cracks in the wetting phase. After five cycles of W/D, the penetration of water into the sample in the wetting phase of the sixth cycle was so great that it showed more strain than moist bentonite by about 7%. Due to the significant reduction of

the $q_u$ and $E_{50}$ values after six cycles of W/D, the toughness value against failure decreased by 85%. According to the stress–strain curves in Figure 6a, the stress of the sample after three and six cycles faced a sharp drop after the failure point compared to the after-curing sample despite the significant increase of the failure strain value. Therefore, the after-curing sample showed significant strength against failure compared to its sample after three and six cycles of W/D. No sudden drop occurred in the stress of the after-curing sample after the failure point, so the failure was far more ductile.

According to Figure 6b, the stress–strain curves of the sample with 8% cement and with an ER/W ratio equal to 1:1 after three and six cycles of W/D dropped significantly. According to Figure 7, the $q_u$ and $E_{50}$ values of this sample decreased by 44 and 66% after three cycles. These values declined by 67 and 89% after six cycles. In addition, the $\varepsilon_f$ value increased by 22 and 84% after three and six cycles, respectively. The toughness value decreased by 56 and 59% after three and six cycles of W/D, respectively, despite the significant increase of the $\varepsilon_f$ values. The considerable decrease in the toughness value was due to the significant decrease of the $q_u$ and $E_{50}$ values. Furthermore, the slight reduction in the sample's toughness by the amount of 3% after six cycles compared to three cycles was due to the increase of the sample's $\varepsilon_f$ by the amount of 52%. This did not indicate that after three additional cycles, the strength of the sample against failure did not change. Thus, it was not evaluated as appropriate.

The drop in the reduction rates of the $q_u$ and $E_{50}$ values of the sample with 8% cement after three and six cycles of W/D had slight changes compared to the sample with 4% cement. By comparing the images of the stabilized samples with 4 and 8% cement in Figures 8 and 9, the amount and width of cracks in the sample with 8% cement were not less than in the sample with 4% cement after similar cycles. Even as shown in Figure 9, the condition of the cracks on the surface of the sample with 8% cement was worse than in the sample with 4% cement. The drop rates in the $q_u$, $E_{50}$, and toughness values of the sample with 12% cement and an ER/W ratio equal to 1:1 compared to the sample with 4% cement after three cycles of W/D decreased by 20, 17, and 9%, respectively. After six cycles of W/D, they declined by 12.5, 6.5, and 13%, respectively. With the addition of 8% more cement (i.e., stabilization with 12% cement), the rates of decrease in the strength parameter values after three and six cycles of W/D compared to the sample with 4% cement had slight changes. As shown in Figures 8 and 10, there were cracks on the surface of the sample with 12% cement during W/D cycles despite adding 8% more cement. The location and type of cracks developed on the surface of the sample with 12% cement during the third to the sixth cycles of W/D differed from the sample with 4% cement. Therefore, the optimum cement content at this concentration of epoxy resin was 4% in the durability process against six successive cycles of W/D.

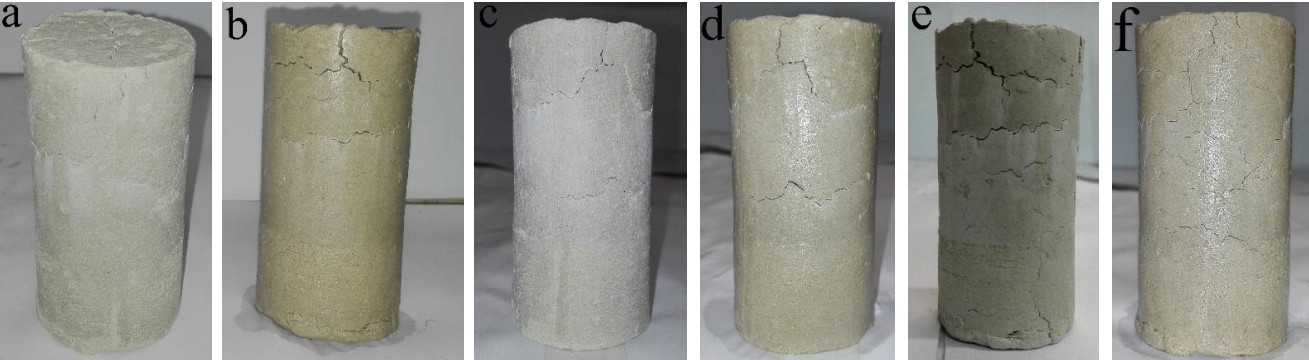

**Figure 9.** The stabilized bentonite sample's with 8% cement and epoxy resin with an ER/W ratio equal to 1:1 at the end of the (**a**) drying of the third cycle; (**b**) wetting of the fourth cycle; (**c**) drying of the fourth cycle; (**d**) wetting of the fifth cycle; (**e**) drying of the fifth cycle; and (**f**) wetting of the sixth cycle.

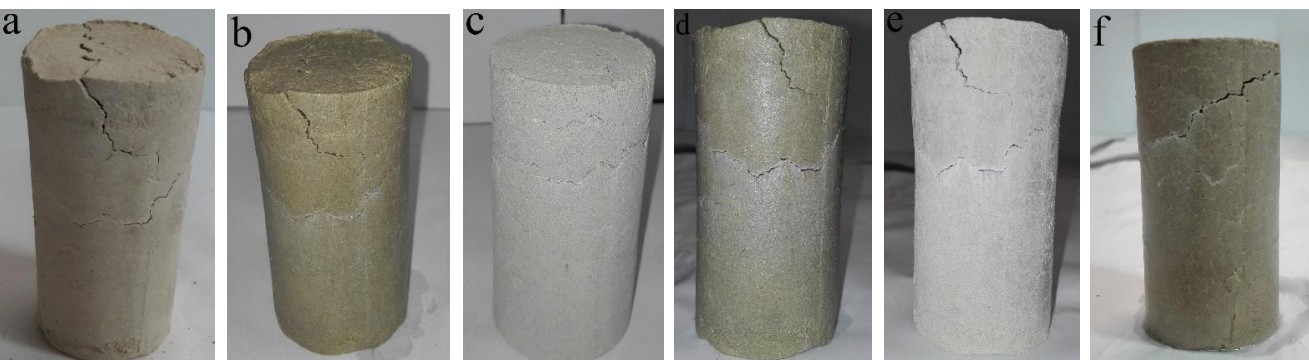

**Figure 10.** The stabilized bentonite sample with 12% cement and epoxy resin with an ER/W ratio equal to 1:1 at the end of the (**a**) drying of the third cycle; (**b**) wetting of the fourth cycle; (**c**) drying of the fourth cycle; (**d**) wetting of the fifth cycle; (**e**) drying of the fifth cycle; and (**f**) wetting of the sixth cycle.

For the initial estimation of the swelling–shrinkage potential of bentonite soil, the durability results of the bentonite samples stabilized with cement and epoxy resin with an ER/W ratio equal to 1:1 were employed in Equations (1) and (2). The estimated coefficients, namely, $C_{qu}$, $C_{E50}$, and $C_{toughnes}$, for the sample with 4% cement were 3, 7, and 6, respectively. For the sample with 12% cement, they were 2.44, 5.1, and 1.85, respectively. To overcome the swelling–shrinkage potential of bentonite soil, the values of $(q_u)_e$, $(E_{50})_e$, and $(toughness)_e$, according to the results of the sample with 4% cement, were estimated to be at least 15, 832, and 1.6 MPa, respectively. According to the results of the sample with 12% cement, they were estimated to be at least 22, 1000, and 1.35 MPa, respectively. Therefore, it was predicted that to overcome the swelling–shrinkage potential of bentonite soil, the additives were needed as stabilizers which increase the $q_u$ value to at least approximately the ultimate compressive strength of OPC concrete.

At an ER/W ratio of 1:1 and for different percentages of cement, the $\varepsilon_f$ value at the end of wetting in the sixth cycle reached approximately 7%. The penetration of water into the sample and the wetting of the bentonite soil caused the stabilized sample, such as the soft wet clay, to have high strain and low strength. According to the stress–strain curves of Figure 6b,c, the failure of the samples stabilized with 8 and 12% cement at the end of wetting in the third and sixth cycles compared to the after-curing samples was brittle despite their significant failure strain values. This was due to their stress dropping at a higher rate after the failure point compared to the after-curing sample. In addition, according to the stress–strain curves of Figure 6, at an ER/W ratio equal to 1:1 and for different percentages of cement, the stress after the failure point in the sixth cycle decreased at a rate approximately equal to that of the third cycle. However, due to the significant increase of the failure strain in the sixth cycle compared to the third cycle, the sample failure at the end of wetting in the sixth cycle was more ductile than the sample at the end of soaking in the third cycle. The creation of more cracks on the sample surface due to three additional cycles and more water penetration into the sample caused more wetting and softening of the soil.

According to Figure 7a, at an ER/W ratio equal to 1:1, the $q_u$ values of the sample with 8% cement after curing and at the end of wetting in the third and sixth cycles were approximately 70% greater than the corresponding values of the sample with 4% cement. The $q_u$ values of the sample with 12% cement at the end of wetting in the third and sixth cycles were 25 and 36% greater than the corresponding values of the sample with 8% cement, respectively. The addition of cement at a content of more than 4% at an ER/W ratio equal to 1:1 did not affect the reduction of the drop rate of the $q_u$ value after three and six cycles of W/D. Still, the optimum cement amount for the samples tested at the end of wetting in both the third and sixth cycles in terms of the $q_u$ parameter was 8%. According to Figure 7b, the toughness values of the sample with 8% cement at the end of wetting in

the third and sixth cycles were 2 and 3.65 times that of the sample with 4% cement. They were 13 and 31.6% greater than in the sample with 12% cement, respectively. Although the increase of the cement content from 8 to 12% had a negligible effect on reducing the drop rate of the toughness value after three and six cycles of W/D, for the toughness parameter to be the same as the $q_u$ parameter at the end of wetting in the third and sixth cycles, the optimum amount of cement was 8%.

According to Figure 7c, at an ER/W ratio of 1:1 and for different percentages of cement, the failure strain value was an ascending function of the number of W/D cycles. Due to the penetration of water into the sample causing softening of the sample, the failure strain value increased with an increasing number of W/D cycles. At the end of wetting in the third cycle, the failure strain value of the sample with 8% cement was 18.1% less than that of the sample with 4% cement, and for the sample with 12% cement, it was 11.3% greater than that of the sample with 8% cement. However, at the end of wetting in the sixth cycle, the addition of cement had little effect on the failure strain values of the samples, and their values were less than 7%. According to Figure 7d, the hardness value of the sample with 8% cement at the end of wetting in the third cycle was 2.36 times that of the sample with 4% cement, and it was 12% less than that of the sample with 12% cement. At the end of wetting in the sixth cycle, the hardness value of the sample with 8% cement was 47% greater than that of the sample with 4% cement, and for the sample with 12% cement, it was 57% greater than that of the sample with 8% cement. The increase of the cement content to more than 4% did not reduce the drop rate of the hardness value of the samples after three and six cycles of W/D. Still, in terms of the hardness parameter at the end of wetting in the third cycle, the optimum amount of cement was 8%. In addition, at the end of soaking in the sixth cycle, the optimum amount of cement was 12%.

- Cement- and epoxy resin-stabilized samples with an ER/W ratio equal to 2:1

According to the stress–strain curves of Figure 6a, despite the slight decrease of the $q_u$ value after three cycles of W/D, the stress behavior of the bentonite sample stabilized with 4% cement and epoxy resin with an ER/W ratio equal to 2:1 was considered more suitable than the after-curing sample. It was unlike the stabilized sample with 4% cement and an ER/W ratio equal to 1:1. At a strain ranging from 0 to about 3.7%, the stress–strain curve of the sample with 4% cement and with an ER/W ratio equal to 2:1 after three cycles of W/D was above that of the after-curing sample. Therefore, the stress increase rate in the first part of the stress–strain curve of its sample at the end of wetting in the third cycle was greater than the corresponding value of the after-curing sample. On the other hand, after three cycles of W/D, in the wide range of the strain near the failure point, its stress was equal to the $q_u$ value. The plastic region expanded at the strain interval with a length of about 3%. The stress drop rate after the failure point for its sample after three cycles of W/D was much lower than in the after-curing sample. In addition, its stress–strain curve dropped after six cycles of W/D and was completely below that of the after-curing sample. However, the magnitude of the reduction was much less than in the sample with an ER/W ratio equal to 1:1.

According to Figure 7, the $q_u$ value of the sample with 4% cement and with an ER/W ratio of 2:1 decreased by 25% after three cycles, and the hardness value not only did not decrease but even increased by 21%. After six cycles, the $q_u$ and $E_{50}$ values decreased by 45 and 50%, respectively. Also, the $\varepsilon_f$ value of this sample increased by the small amount of 8% after three cycles, and the toughness value increased by 20%. After six cycles of W/D, the $\varepsilon_f$ value decreased by 8%, and the toughness value decreased by 53%. For the sample with 4% cement, by doubling the ER/W ratio and increasing it from 1:1 to 2:1, the decrease rates of the $q_u$ and $E_{50}$ values in terms of percentage after three and six cycles of W/D were almost halved.

The development of cracks on the surface of the sample with 4% cement and an ER/W ratio equal to 2:1, from the third cycle of drying to the sixth cycle of wetting, is shown in Figure 11. By comparing the images of this sample in Figure 11 with the stabilized sample with 4% cement and an ER/W ratio equal to 1:1 in Figure 8, it was observed that the

type of crack development on it became different and was not transverse but longitudinal. Furthermore, the amount, depth, and width of cracks developed on it were much less than for the sample with an ER/W ratio equal to 1:1, such that after 24 h of sample immersion and swelling in each cycle, they were not visible. In fact, by doubling the ER/W ratio due to the shallow depth of crack development, the cracks disappeared with the removal the thin crust from the sample surface within 24 h of submerging. No trace of them was seen at the end of the wetting cycle. As shown in Figure 11b,d,f, which are the sample images at the end of wetting in the fourth to the sixth cycles, the crack development created on the sample surface during the drying phase due to the shrinkage of the bentonite soil was not visible after 24 h of wetting. Therefore, by doubling the ER/W ratio, the amount, width, and depth of cracks developed, which were the weaknesses of bentonite samples due to the penetration of water through them into the sample, decreased.

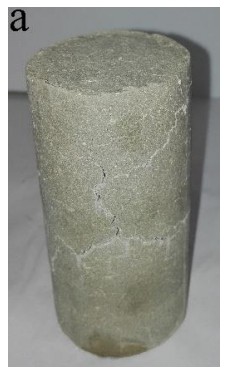 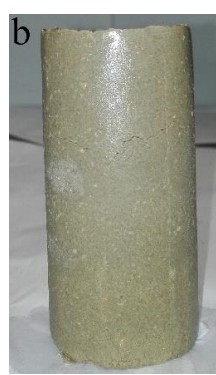 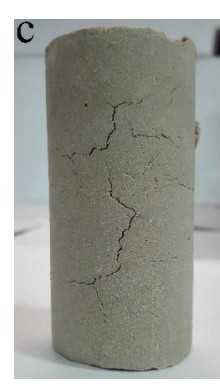 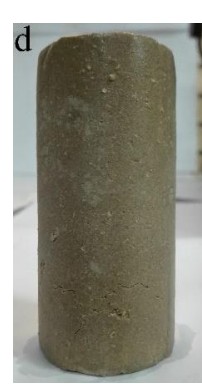 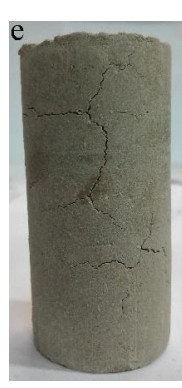 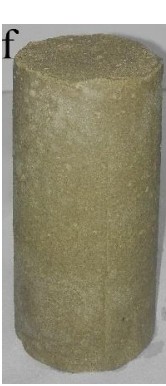

**Figure 11.** The stabilized bentonite sample with 4% cement and epoxy resin with an ER/W ratio equal to 2:1 at the end of the (**a**) drying of the third cycle; (**b**) wetting of the fourth cycle; (**c**) drying of the fourth cycle; (**d**) wetting of the fifth cycle; (**e**) drying of the fifth cycle; and (**f**) wetting of the sixth cycle.

The failure strain value of the stabilized sample with 4% cement and an ER/W ratio equal to 1:1 after six cycles of W/D increased by 61%, while for the stabilized sample with an ER/W ratio equal to 2:1, it decreased by 8%. At an ER/W ratio of 2:1, due to the decrease in the amount, depth, and width of cracks created in the drying phase during the sequence of the W/D cycles, the water penetration into the sample decreased. As a result, the softening of it diminished. So, the sample with an ER/W ratio of 2:1 not only did not have an increase of the $\varepsilon_f$ value at the end of wetting in the sixth cycle, but it decreased.

According to Figure 6b, the stress–strain curve of the bentonite sample stabilized with 8% cement and an ER/W ratio equal to 2:1 dropped after three and six cycles of W/D. After three cycles, it was entirely below the stress–strain curve of the after-curing sample, and after six cycles, it was below the stress–strain curve of the sample subjected to three cycles of W/D. The stress after the failure point for this sample after three and six cycles of W/D demonstrated a significant and sudden drop compared to the after-curing sample. So, the strength against failure decreased significantly after three and six cycles of W/D. As a result, the failure of the sample with 8% cement after three and six cycles of W/D was brittle compared to the after-curing sample. The stresses after the failure point of this sample after three and six cycles of W/D decreased at a rate almost equal to each other. In addition, their $\varepsilon_f$ values were approximately equal. Therefore, the ductility of the sample stabilized with 8% cement did not change after six cycles of W/D compared to the sample subjected to three cycles of W/D. Based on Figure 7, the $q_u$ and $E_{50}$ values of the sample with 8% cement after three cycles declined by 11 and 41%, and they decreased by 32 and 53% after six cycles, respectively. The after-curing $\varepsilon_f$ value of this sample was approximately 6%, and its rate of change after three and six cycles of W/D was negligible. The toughness value of the sample with 8% cement after three and six cycles of W/D decreased by 52 and 61%, respectively.

At an ER/W ratio of 2:1, for the sample with 4% cement after three cycles of W/D, the $q_u$ value decreased by a slight amount, and the stiffness and toughness values of the sample not only did not decrease but improved slightly. The $q_u$ value of the sample with 8% cement decreased slightly after three cycles of W/D, but the stiffness and toughness values of the sample suffered relatively significant reductions. The decrease rates of the strength parameter values of the sample with 8% cement after six successive cycles of W/D compared to the sample with 4% cement showed slight changes. Therefore, adding 4% more cement (i.e., stabilization with 8% cement) had a negligible effect on reducing the drop rate of the strength parameter values of the stabilized sample with an ER/W ratio of 2:1 regarding durability against W/D cycles. By comparing the images of the samples stabilized with 4 and 8% cement after three and six cycles of W/D in Figures 11 and 12, despite the difference in the crack development on these samples, it was observed that the width and depth of cracks on the surface of the sample with 8% cement were not less than for the sample with 4% cement after similar cycles. As shown in Figure 12e, the crack development on the surface of the sample with 8% cement was more critical than on the sample with 4% cement shown in Figure 11e. It should be noted that the development of cracks on the sample shown in Figure 12e was superficial. The cracks were not deep, such that after 24 h of sample wetting in the sixth cycle and the falling of the thin shell off its surface, no traces of them were seen (Figure 12f).

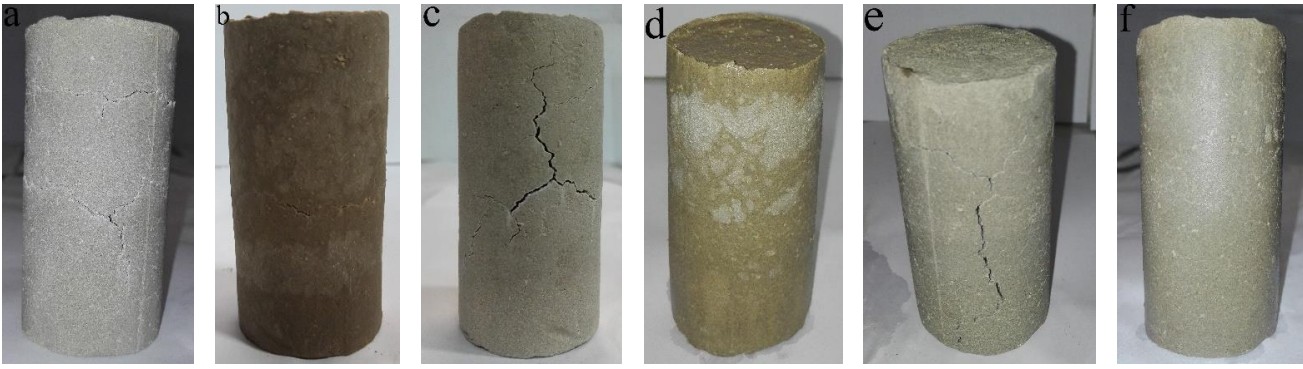

**Figure 12.** The stabilized bentonite sample with 8% cement and epoxy resin with an ER/W ratio equal to 2:1 at the end of the (**a**) drying of the third cycle; (**b**) wetting of the fourth cycle; (**c**) drying of the fourth cycle; (**d**) wetting of the fifth cycle; (**e**) drying of the fifth cycle; and (**f**) wetting of the sixth cycle.

According to Figure 6c, the stress–strain curves of the bentonite sample stabilized with 12% cement and an ER/W ratio equal to 2:1 dropped to some extent after three and six successive cycles of W/D. Both were entirely below the after-curing stress–strain curve. The first parts of the stress–strain curves of the samples, after three and six cycles of W/D, were at a slight distance from each other. Even at the strain ranging from 0 to approximately 4.4%, the stress–strain curve of the sample after six cycles was on top of that of the sample after three cycles with a slight distance. The stress of the sample subjected to three successive cycles of W/D after the failure point encountered a significant and sudden drop compared to the after-curing sample. Therefore, its failure was brittle, and it had less strength against failure than the after-curing sample. In addition, the stress after the failure point for the sample after three cycles faced a sudden drop compared to the sample after six cycles of W/D. Therefore, its failure relative to the sample subjected to six cycles was also brittle. According to Figure 7, the $q_u$ and $E_{50}$ values of the sample with 12% cement after three cycles decreased by 4 and 37%. After six cycles, they declined by 10.1 and 33.2%, respectively. The changes of the $\varepsilon_f$ values after three and six cycles of W/D were less than 10%. The toughness value of the sample decreased by 55% after three cycles and decreased by 27.5% after six cycles. Due to the ductile failure of the sample subjected to six cycles of

W/D compared to the sample subjected to the three cycles, its toughness value was far greater than that of the sample after three cycles.

By adding 12% cement at an ER/W ratio equal to 2:1, the reductions in the drop rates of the $q_u$ and $E_{50}$ values after three cycles of W/D compared to the sample with 8% cement were 60 and 11.5%, respectively. After six cycles of W/D, they were 68 and 37%, respectively. The reduction in the drop rate of the toughness value of this sample after three cycles of W/D compared to the sample with 8% cement was negligible, but after six cycles, it was 55%. Therefore, by adding 12% cement, the drop rate in the strength parameter values of the sample decreased after three and six cycles of W/D compared to the sample with 8% cement. Adding 12% cement at this epoxy resin concentration was appropriate. The images of the bentonite sample stabilized with 12% cement and an ER/W ratio equal to 2:1, from the third cycle of drying to the sixth cycle of wetting, are shown in Figure 13. By comparing them with the images of the stabilized samples with 4 and 8% cement in Figures 11 and 12, it was observed that the development of the cracks on the surface of the sample with 12% cement at the drying phase of the third, fourth, and fifth cycles, unlike the samples with 4 and 8% cement, significantly decreased, such that they were not clearly visible. Therefore, at an ER/W ratio equal to 2:1, the addition of 12% cement was very effective in reducing the width and depth of cracks. Still, the coherence of the sample was not sufficient not to create crack development at the drying phase of each cycle.

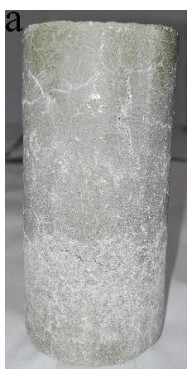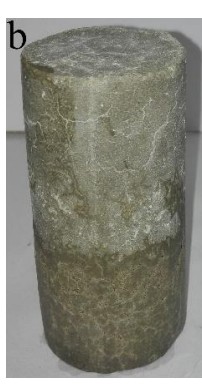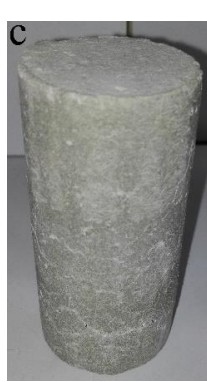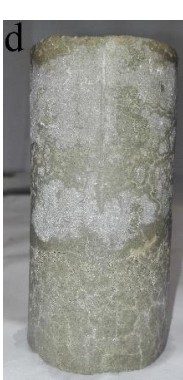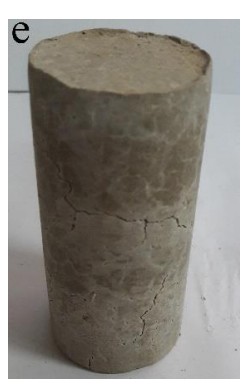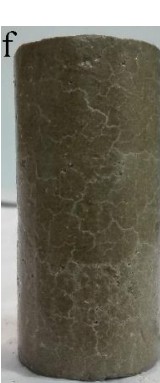

**Figure 13.** The stabilized bentonite sample with 12% cement and epoxy resin with an ER/W ratio equal to 2:1 at the end of the (**a**) drying of the third cycle; (**b**) wetting of the fourth cycle; (**c**) drying of the fourth cycle; (**d**) wetting of the fifth cycle; (**e**) drying of the fifth cycle; and (**f**) wetting of the sixth cycle.

For another estimation of the swelling–shrinkage potential of bentonite soil, the durability results of the bentonite samples stabilized with cement and epoxy resin at an ER/W ratio equal to 2:1 were employed in Equations (1) and (2). The estimated coefficients, namely, $C_{qu}$, $C_{E50}$, and $C_{toughnes}$, for the sample with 4% cement were 1.82, 2, and 2.12, respectively, and for the sample with 8% cement, they were determined to be 1.47, 2.1, and 2.57, respectively. For the sample with 12% cement, they were calculated to be 1.11, 1.5, and 1.38, respectively. To overcome the swelling–shrinkage potential of bentonite soil, the values of $(q_u)_e$, $(E_{50})_e$, and $(toughness)_e$, according to the results of the sample with 4% cement, were estimated to be at least 25.4, 422, and 14 MPa respectively. According to the sample results with 8% cement, they were predicted to be at least 18.8, 571, and 2.2 MPa, respectively, and based on the sample results with 12% cement, were estimated to be at least 13.9, 427, and 1.24 MPa, respectively. The required values of $q_u$, hardness, and toughness after stabilization using any method to overcome the swelling–shrinkage potential of bentonite soil were estimated to be at least 25.4, 571, and 2.2 MPa, respectively.

Based on Figure 7a, at an ER/W ratio equal to 2:1, the $q_u$ values of the samples with 8 and 12% cement were 10 and 15% more than the sample with 4% cement at the end of wetting in the third cycle, respectively. These increases of the $q_u$ values were made negligible by adding 4 and 8% more cement. The after-curing $q_u$ value of the sample with

4% cement was greater than those of the samples with 8 and 12% cement. So, both after curing and after the third cycle, adding cement in excess of 4% had little effect on the $q_u$ value of the sample. At the end of wetting in the sixth cycle, the $q_u$ values of the stabilized samples with 8 and 12% cement were 29 and 47% more than those of the samples with 4% cement, respectively. After three additional cycles (i.e., at the end of wetting in the sixth cycle), the addition of 12% cement had some effect on improving the $q_u$ value. Therefore, for the $q_u$ parameter of the samples subjected to six cycles of W/D, the optimum amount of cement was 12%.

According to Figure 7b, at an ER/W ratio equal to 2:1, the toughness value of the sample with 8% cement was 48% less than that of the sample with 4% cement, and for the sample with 12% cement, it was 1% less than that of the sample with 8% cement at the end of wetting in the third cycle. At the end of wetting in the third cycle, the reduction in the toughness value of the sample with 8% cement was due to the decrease in the $\varepsilon_f$ value by 15%. Therefore, it could not be inferred that increasing the cement percentage had reduced the sample strength against failure. At the end of wetting in the sixth cycle, the toughness of the sample with 8% cement was by a slight amount of 7% greater than the sample with 4% cement. Still, for the sample with 12% cement, it was almost four times that of the sample with 4% cement. The $\varepsilon_f$ value of the sample in the sixth cycle had slight changes with increasing cement percentage. After the three additional cycles (i.e., at the end of wetting in the sixth cycle), adding cement in the amount of 12% significantly affected the toughness value of the sample against failure. As a result, for the toughness parameter of the stabilized sample subjected to six cycles of W/D, the optimum cement amount was 12%.

According to Figure 7c, at the end of wetting in the third cycle, the failure strain value of the stabilized sample with 8% cement and with an ER/W ratio of 2:1 decreased by 15% compared to the sample with 4% cement. By adding 12% cement, it decreased by 3% compared to the sample with 8% cement. At the end of wetting in the sixth cycle, the addition of cement in excess of 4% had almost no effect on the failure strain value, and its changes were negligible. According to Figure 7d, at the end of wetting in the third cycle, the hardness values of the samples with 8 and 12% cement were 37 and 29% less than that of the sample with 4% cement, respectively. At the end of wetting in the sixth cycle, the hardness of the sample with 8% cement by the slight amount of 21% and the hardness of the sample with 12% cement by the significant amount of 78% were greater than that of the sample with 4% cement. Finally, according to Figure 7, at an ER/W ratio equal to 2:1, the optimum amount of cement was 12%, which improved the strength parameters of the bentonite sample in terms of durability for up to six cycles of W/D.

*3.4. Evaluation of Epoxy Resin-Stabilized Bentonite Soil without Cement and Water*

In this part, the stabilization of the bentonite soil sample was performed such that the total amount of the optimum moisture content required for the compaction was replaced with epoxy resin additive. The samples were then treated for 7 days under the same conditions as the other stabilized bentonite samples. The purpose of stabilizing bentonite soil with only epoxy resin additive without cement and water was to increase the strength parameter values to the level of about OPC concrete. Furthermore, if this occurred, the swelling–shrinkage potential of bentonite soil was checked by exposing the epoxy resin-stabilized sample to six successive cycles of W/D, as in the previous sections. The results of uniaxial tests performed on the samples after curing and at the end of wetting in the third and sixth cycles are given in Figure 14. Their stress–strain curves are shown in Figure 14a. The changes in the values of their parameters for $q_u$, toughness, $E_{50}$, and $\varepsilon_f$ are presented in Figure 14b,c,d,e, respectively.

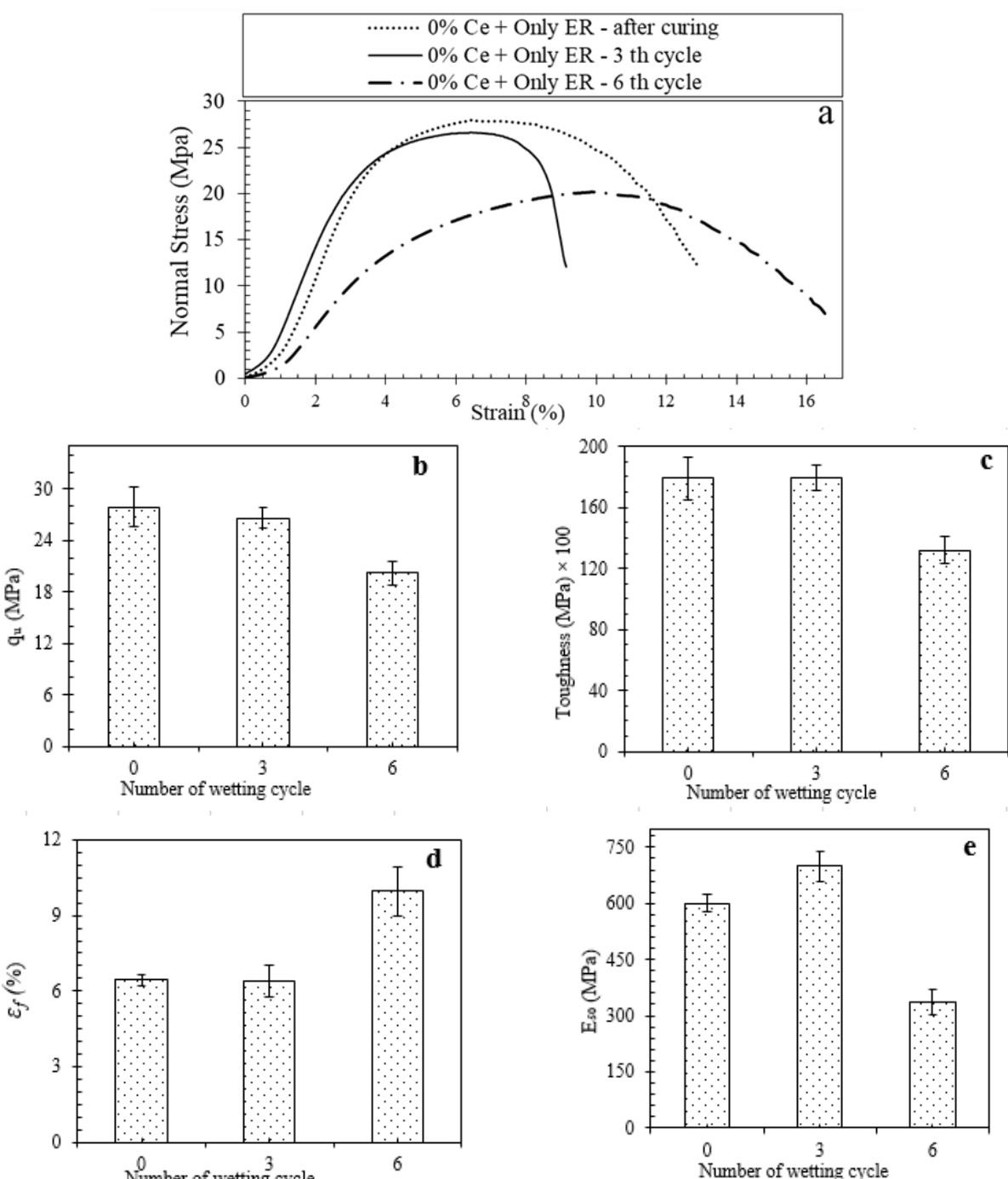

**Figure 14.** (**a**) The stress–strain curves; the changes of (**b**) qu; (**c**) toughness; (**d**) failure strain; and (**e**) secant elastic modulus of stabilized bentonite soil with only epoxy resin and no cement and water, after curing and at the end of wetting in the third and sixth cycles.

### 3.4.1. Evaluation of the Strength Parameters after 7 Days of Curing

According to the stress–strain curve of Figure 14a, the stress of the stabilized bentonite sample with only epoxy resin at the strain interval with a length of approximately 5%, which was a wide strain range, was almost equal to $q_u$. As a result, the plastic region expanded over a wide strain range with a length of approximately 5%, in which the difference between the sample stress and the $q_u$ value was negligible. Therefore, it showed a lot of ductility, toughness, and strength against failure. Its $q_u$ and failure strain values were 28 MPa and 6.44%, respectively. After the performance, the dry unit density and

strength are two important properties of lightweight structural concrete. The advantages of such materials are the high ratio of strength to dry unit density and the low cost of concrete. Usually, in a mixing plan, the 28-day compressive strength of OPC concrete is in the range of 20 to 35 MPa, and the dry unit density of most lightweight structural concrete is between 1600 and 1760 kg/m$^3$. Therefore, the $q_u$ value of bentonite soil stabilized with only epoxy resin and no cement and water was in the range of OPC concrete, while the failure strain value was approximately 32 times that of OPC concrete. In addition, its dry unit density was about 70% of that of concrete. Therefore, the bentonite sample stabilized with only epoxy resin, in addition to having high strength and ductility, was much lighter than OPC concrete. It was consequently cost-effective from the strength and economic points of view.

At an ER/W ratio of 2:1, the optimum amount of cement was 12%. By adding epoxy resin in the amount of the optimum moisture, the $q_u$, hardness, and toughness values became two times that of this sample. The amount of change in its failure strain compared to the sample with 12% cement was small and less than 11%. For the sample with an ER/W ratio of 2:1, the strain interval length in which the plastic region expanded was approximately 3%. In contrast, for the stabilized sample with only epoxy resin, it was about 5%. Therefore, the failure of both samples was ductile, but the ductility and the strength against failure of the sample stabilized with only epoxy resin were much higher.

3.4.2. Evaluation of Durability against W/D Cycles

According to Figure 14a, the stress–strain curve of the bentonite sample stabilized with only epoxy resin at the end of the wetting in the third cycle did not decrease in the wide strain ranging from 0 to approximately 3.5%. Rather, it was on top of that of the after-curing sample, at a slight distance. The plastic region in which the stress was in the range of the $q_u$ value expanded at the strain interval with a length of about 3%. However, for the after-curing sample, it was approximately 5%. Therefore, after three successive cycles of W/D, the weakness that the stress–strain curve indicated that had occurred was the reduction in the toughness of the sample against failure. According to Figure 14b,c,d,e, the $q_u$ value of this sample decreased by a tiny amount of 5% after three cycles of W/D. The failure strain value had slight changes, but the toughness value against failure decreased by 31.5%. Its hardness value not only did not decrease but even increased by 16.3%. It could be concluded that the sample maintained its rigidity and strength after three cycles of W/D.

According to Figure 14a, the stress–strain curve of the bentonite sample stabilized with only epoxy resin at the end of wetting in the sixth cycle dropped significantly over a wide strain range. Along with this, it showed very high ductility. According to Figure 14b–e, at the end of wetting in the sixth cycle, the $q_u$, hardness, and toughness values of this sample decreased by 27.6, 44, and 11%. The failure strain value increased by 55% and gained approximately 10%. The stress drop of the sample proceeded at a low rate after the failure point. The increase of the failure strain value was significant. Although, after six cycles of W/D, the bentonite sample stabilized with only epoxy resin showed very considerable ductility, the stabilization did not have the expected efficiency due to the significant decrease of the strength parameter values of the sample.

The images of the bentonite sample stabilized with only epoxy resin from the third cycle of drying to the sixth cycle of wetting are shown in Figure 15. At the end of drying in the third cycle to the fifth cycle, no crack development was observed on the sample surface. So, to overcome the shrinkage potential of bentonite soil and prevent crack development on the sample surface during the drying phase of each cycle, the strength parameter values of the sample should be at least equal to those of OPC concrete.

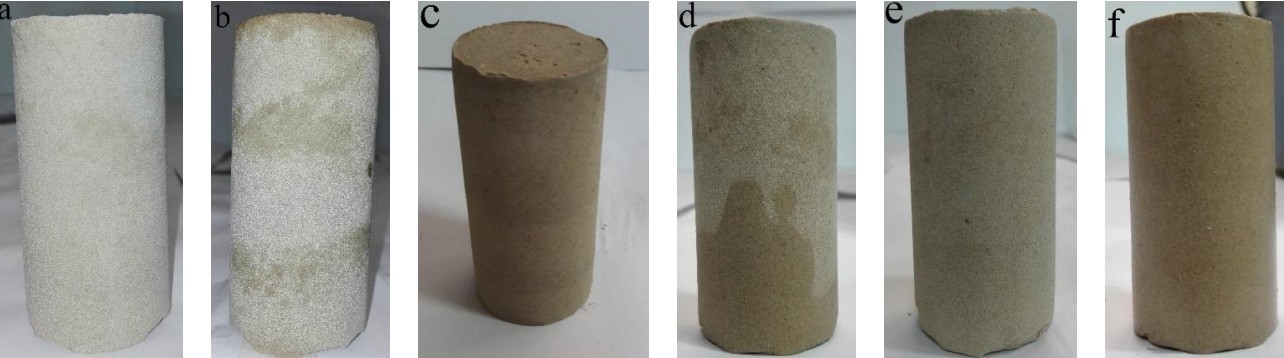

**Figure 15.** The stabilized bentonite sample with only epoxy resin and no cement and water at the end of the (**a**) drying of the third cycle; (**b**) wetting of the fourth cycle; (**c**) drying of the fourth cycle; (**d**) wetting of the fifth cycle; (**e**) drying of the fifth cycle; and (**f**) wetting of the sixth cycle.

The loss rates of the $q_u$ and hardness values for bentonite samples stabilized with only epoxy resin after six cycles of W/D were 27.6 and 44%, respectively, and they did not decrease compared to the sample stabilized with 12% cement and an ER/W ratio equal to 2:1. Although no visible crack development was seen on the surface of the stabilized bentonite sample with only epoxy resin, it was not yet wholly impenetrable.

Therefore, to make the bentonite soil impenetrable against the swelling–shrinkage potential, additives were needed that increased the stiffness and strength to values higher than those of OPC concrete. The estimated coefficients, namely, $C_{qu}$, $C_{E50}$, and $C_{toughnes}$, for the stabilized bentonite sample with only epoxy resin were calculated to be 1.4, 1.8, and 1.36, respectively. In order to render bentonite soil impenetrable against wetting in each cycle and thoroughly overcome the swelling–shrinkage potential, it was estimated that its values of $(q_u)_e$, $(E_{50})_e$, and $(toughness)_e$ after stabilization using any method should reach at least 38.6, 1074, and 2.45 MPa, respectively.

*3.5. Geoenvironmental Assessment of Clay Plastic Concrete and Progress in Sustainability*

Almost all civil and road construction projects are performed on soil. Land development and increasing population growth have led to the reclamation and reuse of lands with poor mechanical properties from the viewpoint of geoenvironmental engineering. The deep mixing method (DMM) is used in many geotechnical projects to reduce the settlement and increase the bearing capacity of weak soils under the foundations of structures [72].

In the DMM, binders such as cement and lime are mixed with soil to form strong stone columns [25]. According to the results of this research and other studies [25,49,65,67,69–71], mixing cement with soft clay soils, due to the effect of the clay minerals' type, could not significantly increase the strength and durability of the soil-and-cement mixture. This weakness has caused the construction of deep foundations, such as piles, to be preferred to the DMM and stone columns [23].

Due to the high volume of cement used in the construction of foundations such as piles, mixing soft clay soils using the DMM to form stone columns with other materials that could remarkably increase the strength and ductility of stabilized soft clay soils has many advantages, especially from the viewpoint of the environment. One of the valuable benefits of stabilization with epoxy resin is its great efficiency in soft clay soils containing high percentages of the mineral montmorillonite.

Epoxy resin, in addition to its great performance in stabilizing clay soils, in combination with clay soil, could be a good substitute for both cement and water. This would save water consumption and protect the environment from another point of view. Clay plastic concrete, including bentonite and epoxy resin, being improved by the DMM and stone columns compared to concrete piles, contributes to progress in sustainable construction.

## 4. Conclusions

The swelling–shrinkage potential of bentonite soil was so high that it lasted for fewer than two cycles of W/D despite stabilization with 30% cement and curing for 28 days. Another additive as a stabilizer was needed to meet the durability criteria. Therefore, the addition of epoxy resin with different ER/W ratios to cement-stabilized bentonite soil was investigated. The following conclusions can be specifically drawn from the present study:

1. By adding epoxy resin at ER/W ratios equal to 1:4 and 1:2 and for different percentages of cement, the ductility of the samples improved noticeably relative to the samples stabilized with cement and without epoxy resin. Although at these ER/W ratios of less than 1:1, the stabilized samples had better performance in durability by adding cement, they eventually lasted for up to two cycles.

2. The minimum ER/W ratio for different percentages of cement in bentonite soil samples that lasted for up to six cycles of W/D was equal to 1:1. However, at this ratio, the W/D cycles had devastating effects on the samples. The development of cracks on the samples' surfaces during drying and water penetration into the samples during the wetting in each cycle caused their strength parameter values after six cycles of W/D to drop by more than 70%. The optimum amount of cement for them was 8%.

3. At an ER/W ratio of 1:1 and for different percentages of cement, the after-curing failure strain value was approximately 4%. At the end of wetting in the sixth cycle, it reached about 7%. At an ER/W ratio of 2:1, the after-curing failure strain was approximately 6%, while its value changed slightly after wetting in the sixth cycle. By doubling the ER/W ratio, an extensive plastic region in the stress–strain curve was achieved at the strain interval with a length of approximately 3%. The ductility, stiffness, and strength values of the samples increased so that the effects of W/D cycles on them considerably decreased.

4. Increasing the cement content at an ER/W ratio equal to 2:1 had significant effects on improving the performance of the samples in the durability process, and the optimum amount of cement was 12%.

5. The $q_u$ value of bentonite soil stabilized with only epoxy resin and without cement and water reached the range of OPC concrete. At the same time, the failure strain value became 32 times greater, and the plastic region of the stress–strain curve expanded over a wide range of strain with a length of approximately 5%. Its ductility and toughness against failure were substantially higher than those of OPC concrete, which was much lighter. The shrinkage potential of bentonite soil was overcome by this stabilization, and no crack development was seen in the drying phase of each cycle on the sample surface. However, the strength parameters still decreased after six cycles of W/D.

6. In order to make the bentonite soil samples completely impenetrable and overcome the swelling potential in the wetting phase of each cycle, it was predicted that the $(q_u)_e$, $(E_{50})_e$, and $(\text{toughness})_e$ values of bentonite soil after stabilization should reach at least 38.6, 1074, and 2.45 MPa, respectively. It was necessary to increase the ductility and strength to values much greater than those of OPC concrete. Thus, the strength parameters do not decrease during the wetting phase in each cycle.

**Author Contributions:** Conceptualization, S.S. and S.M.M.; methodology, S.S.; validation, S.S., S.M.M. and N.B.; formal analysis, S.S.; investigation, S.S. and S.M.M.; resources, S.S.; data curation, S.S.; writing—original draft preparation, S.S.; writing—review and editing, S.S., S.M.M. and N.B.; visualization, S.S.; supervision, S.M.M.; project administration, S.M.M.; All authors have read and agreed to the published version of the manuscript.

**Funding:** This research received no external funding.

**Institutional Review Board Statement:** Not applicable.

**Informed Consent Statement:** Not applicable.

**Data Availability Statement:** The authors confirm that the data supporting the findings of this study are available within the article.

**Conflicts of Interest:** The authors declare no conflict of interest.

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
