# Peer review of "Performance Evaluation of Clay Plastic Concrete of Cement and Epoxy Resin Composite as a Sustainable Construction Material in the Durability Process"

_sustainability, doi:10.3390/su15118987_

Round 1

Reviewer 1 Report

This paper shows an investigation on the performance evaluation of clay plastic concrete of cement and epoxy resin composite as a sustainable construction material in the durability process. Generally, this topic is interesting, and this paper can be considered after the following comments being addressed.

(1) The background of this work should be highlighted in abstract section.

(2) The abbreviation appeared at the first time should be clear in abstract section. Such as W/D.

(3) The cited references “[1-2]” should be corrected into “[1,2]”, and the related problems should be checked and corrected throughout the whole manuscript.

(4) The author should give a review on the performance of sustainable cement-based materials. The following to references may be helpful for this review: “https://doi.org/10.1016/j.conbuildmat.2023.131328” “https://doi.org/10.1016/j.cemconcomp.2022.104629”

(5) The figures at the current form are not clear, which should be replaced in the revised manuscript.

(6) The error bar should be given in Fig. 14.

(7) The conclusion section is too long, which should be shortened in the revised manuscript.

 Moderate editing of English language

Author Response

We would like to thank the Reviewers for taking the time and effort necessary to review the manuscript. We sincerely appreciate all valuable comments and suggestions, which helped us to improve the quality of the manuscript. The responses, and explanations related to their comments are enclosed below.

Reviewer 2 Report

This manuscript presents a very interesting research topic. The authors evaluated the durability of bentonite soil stabilized with cement and epoxy resin additives, increasing gradually the ratio of epoxy resin. The results showed that concrete with epoxy resin presents strength and rigidity similar to conventional concrete.

 Before the article is accepted for publication, I suggest authors do a literature review on more recent articles/literature. And, to be attractive to readers, authors should highlight how "sustainability" is associated with the article. And other minor suggestions are described below:

1. Most articles in the introduction are more than 5 years old, check if there are more current articles to improve the state of the art.

2. Table 1 does not exist in the article; it starts with Table 2. It is necessary to check the sequence of tables and also make changes to the text.

3. Caption of Figure 2 is incomplete

4. Figure 1 does not exist in the article, need to add or edit captions for current figures and text.

5. The results and discussions have few discussion based on current literature and comparisons among other papers.

6. I suggest the inclusion of standard deviation in the figures.

7. The conclusion section, in my opinion it is too extensive, it should be more elaborate based on the main results. 

I think English is fine

Author Response

(The authors gave the same response as above.)

Reviewer 3 Report

The present paper deals with a possible strategy of Bentonite soil stabilization taking in account a mixture containing cement and epoxy resin. The topic is interesting and appealing for the reader but improvments are needed for publication.

In the abstract:

-please avoid acronym without explanation such as W/D;

-Line 14, unixial test of what property?

-line 18, if is a ratio, it can't be 1, but 1:1, please correct in all the document;

-line 22, these results are referred to...what? 

-keywords, maybe are too many, please check carefully the authors instructions;

In the Introduction, the last paragraph (line 113 to 125) is not clear as the topic of the present paper is explained but it is not explicit at that point. Furhermore, more reference to previous work regarding Bentonite should  be included. In the same paragraph anticipation of the results must be avoided. 

In the materials and methods :

- the first table is named table 2, please correct;

- please revise line 130 as in table 2 the XRD analysis results are reported, also XRD pattern should be showed. 

- line 141 to 148 and line 152 to 158 must be moved to the result section

- line 175 please clarify the sentence starting with "This homogeneous.."

- A Table for summarize all the formulation tested should be shown and code should be applied and employed in the discussion to sake of clariness to the reader;

- Figure 2, please reconsider the caption of the figure as there is some text 

- paragraph 3 must be included in 2.3 as it has no sense separated by it. 

- paragraph 4.1 is rendountant with the introdutcion, 

- Error bars must be indicated and discussed in Figure 3b, c, d, e

- line 312 to 323 must be moved to the introductin

- Error bars must be indicated and discussed in Figure 4 c, d, e, f

- Error bars must be indicated and discussed in Figure 7

- Error bars must be indicated and discussed in Figure 14b, c, d, 

In the discussion section the  evaluation of the  environmental impact of epoxy resin in the soil should be discussed 

none

Author Response

(The authors gave the same response as above.)

Round 2

Reviewer 3 Report

All the comments have been satisfied and the paper is now suitable for publication.